# STAR-BENCH: PROBING DEEP SPATIO-TEMPORAL REASONING AS AUDIO 4D INTELLIGENCE

**Zihan Liu**[1,2*], **Zhikang Niu**[3,5*], **Qiuyang Xiao**[3], **Zhisheng Zheng**[3], **Ruoqi Yuan**[1], **Yuhang Zang**[2✉],
**Yuhang Cao**[2], **Xiaoyi Dong**[2,4], **Jianze Liang**[2], **Xie Chen**[3,5], **Leilei Sun**[1], **Dahua Lin**[2,4], **Jiaqi Wang**[2,5✉]
[1] Beihang University, [2] Shanghai AI Laboratory, [3] Shanghai Jiao Tong University,
[4]The Chinese University of Hong Kong, [5] Shanghai Innovation Institute
liuzihan@buaa.edu.cn, zangyuhang@pjlab.org.cn
**Code:** https://github.com/InternLM/StarBench
**Benchmark:** https://huggingface.co/datasets/internlm/STAR-Bench
**Homepage:** https://internlm.github.io/StarBench

## ABSTRACT

Despite rapid progress in Multi-modal Large Language Models and Large Audio-Language Models, existing audio benchmarks largely test semantics that can be recovered from text captions, masking deficits in fine-grained perceptual reasoning. We formalize audio **4D intelligence** that is defined as reasoning over sound dynamics in time and 3D space, and introduce **STAR-Bench** to measure it. STAR-Bench combines a Foundational Acoustic Perception setting (six attributes under absolute and relative regimes) with a Holistic Spatio-Temporal Reasoning setting that includes segment reordering for continuous and discrete processes and spatial tasks spanning static localization, multi-source relations, and dynamic trajectories. Our data curation pipeline uses two methods to ensure high-quality samples. For foundational tasks, we use procedurally synthesized and physics-simulated audio. For holistic data, we follow a four-stage process that includes human annotation and final selection based on human performance. Unlike prior benchmarks where caption-only answering reduces accuracy slightly, STAR-Bench induces far larger drops (-31.5% temporal, -35.2% spatial), evidencing its focus on linguistically hard-to-describe cues. Evaluating 19 models reveals substantial gaps compared with humans and a capability hierarchy: closed-source models are bottlenecked by fine-grained perception, while open-source models lag across perception, knowledge, and reasoning. Our STAR-Bench provides critical insights and a clear path forward for developing future models with a more robust understanding of the physical world.

## 1 INTRODUCTION

As a fundamental modality of human perception, audio serves a pivotal role in communication, aesthetic appreciation, and situational awareness, complementing the limitations of visual perception. With the rise of Multimodal Large Language Models (MLLMs) (Comanici et al., 2025; Achiam et al., 2023) and especially Large Audio-Language Models (LALMs) (Chu et al., 2024; Goel et al., 2025), these models have shown impressive capabilities in understanding audio, representing a crucial step toward diverse applications such as embodied intelligence (Paul et al., 2022).

To drive progress, a series of audio benchmarks has been introduced (Yang et al., 2024; Sakshi et al., 2025), covering traditional tasks like Automatic Speech Recognition (ASR) and sound event classification. While some recent efforts are beginning to emphasize reasoning abilities (Ma et al., 2025; Kumar et al., 2025), we observe that existing benchmarks predominantly focus on coarse-grained semantic content, which is audio information that can be distilled into textual descriptions with minimal loss. As shown in the **left** part of Fig. 1, we first use Gemini 2.5 Pro (Comanici et al., 2025) to generate detailed audio captions for samples in recent representative audio benchmarks MMAU (test-mini) (Sakshi et al., 2025) and MMAR (Ma et al., 2025). We then prompt the model to answer questions based *only* on these audio captions, and its performance drops by only 5.9% and 9.0%,

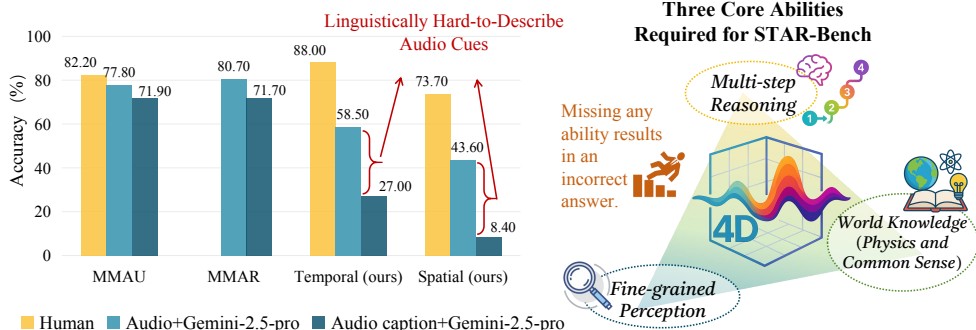

Figure 1: **(Left)**: A comparison between humans and the Gemini 2.5 Pro with and without audio captions on various audio benchmarks. Our STAR-Bench evaluates linguistically hard-to-describe audio cues. See Appendix B.1 for audio caption details. **(Right)**: The three core abilities required to solve tasks in the STAR-Bench benchmark.

respectively, compared to when it processes the raw audio. This result suggests that existing benchmarks primarily evaluate audio information that is **easily representable by text**. However, human auditory intelligence is not limited to this coarse-grained understanding. For example, humans can intuitively judge the water level in a container from the dynamic changes in the pouring sound, even without being able to precisely articulate the underlying acoustic features. Similarly, we can infer the trajectory and distance of a vehicle approaching from behind to ensure our safety. These abilities are rooted in deep reasoning of audio cues **that are difficult to represent linguistically**.

To capture this human-like audio competence, we propose a new paradigm, called **audio 4D intelligence**. This is defined as the ability to perform deep reasoning over the dynamics of **sound sources** in **time (1D)** and **three-dimensional space (3D)**, grounded in an understanding of the physical world. Mastering 4D audio intelligence is crucial for various applications. In embodied AI and robotics, for instance, agents must integrate fine-grained auditory cues to interact naturally with their surroundings, such as using sound to infer the trajectory of an object or to monitor the subtle operations of a machine. To systematically evaluate this paradigm and bridge the gap between current audio benchmarks and real-world auditory intelligence, we introduce the **S**patio-**T**emporal **A**udio **R**easoning (**STAR-BENCH**) benchmark.

STAR-BENCH is designed through a hierarchical task structure with two levels. At the **Foundational Acoustic Perception** level, we conduct a fine-grained, quantitative evaluation of six core audio attributes (pitch, loudness, duration, azimuth, elevation, distance) across both absolute perception ranges and relative discrimination sensitivity. We also introduce a **Holistic Spatio-Temporal Reasoning** level that evaluates an audio model's ability to infer both event order and 3D scene structure. Temporal reasoning is tested via segment reordering that spans continuous processes and discrete event scripts, while spatial reasoning covers static localization, multi-source relations, and dynamic trajectory tracking. As shown in the **right** part of Fig. 1, every question in our holistic tasks is designed to probe a synthesis of three core pillars, such as multi-step reasoning. A failure in any one of these pillars will lead to an incorrect response. Our **data curation pipeline** couples procedurally synthesized, fully parameterized audio for foundational perception with large-scale real-world corpora for holistic reasoning. For the latter, we use a four-stage process including **human annotation** and **final selection by human performance** to ensure the high quality of benchmark samples.

Our comprehensive evaluation of 19 models (16 open-source and 3 closed-source) reveals a clear capability hierarchy between the two groups. Leading closed-source models like Gemini 2.5 Pro excel in knowledge and reasoning, shifting their primary bottleneck to the more difficult challenge of fine-grained perception. In contrast, open-source models exhibit fundamental weaknesses across all three core capabilities. Through our detailed error analysis and ablation studies, we highlight several key insights for the future development of open-source audio models: 1) **Enhancing dense audio captioning.** Open-source models struggle to produce dense, fine-grained captions, which limits their perceptual sensitivity and ability to extract embedded knowledge. Bridging this gap is a crucial first step. 2) **Improving multi-audio reasoning.** Open-source models lag significantly in comparing, integrating, and grounding information across multiple audio clips. 3) **Moving beyond channel-averaged audio preprocessing.** The common practice of averaging multi-channel audio

Table 1: A comparative overview of our benchmark against other representative audio benchmarks. (✓: Fully supported, ◖: Partially supported or limited amount, ✗: Not supported)

| Benchmark | Temporal Deep Reasoning | Spatial Deep Reasoning | Quantitative Attribute Evaluation | Robust Evaluation | Multi-Audio | Fully Human-Annotated | Fully Expert Verified |
|---|---|---|---|---|---|---|---|
| AIR-Bench [45] | ✗ | ✗ | ✗ | ✗ | ✗ | ✗ | ✗ |
| MMAU [30] | ✗ | ✗ | ✗ | ✗ | ✗ | ✓ | ✓ |
| Dynamic-SUPERB Phase-2 [16] | ✗ | ✗ | ✗ | ✗ | ◖ | ◖ | ✗ |
| MMAR [27] | ✗ | ◖ | ✗ | ✗ | ◖ | ✓ | ✓ |
| MMAU-Pro [20] | ✗ | ◖ | ✗ | ✗ | ✓ | ✓ | ✓ |
| **STAR-BENCH (ours)** | ✓ | ✓ | ✓ | ✓ | ✓ | ✓ | ✓ |

into a mono signal is a major bottleneck for spatial reasoning. Developing architectures that natively process multi-channel cues is essential for unlocking genuine spatial awareness.

Our contributions are summarized as: **(1)** We formalize **audio 4D intelligence**, and empirically show that prior benchmarks largely probe text-representable semantics, motivating a shift toward fine-grained, non-linguistic auditory cues. **(2)** We introduce the STAR-BENCH with foundational acoustic perception and holistic spatio-temporal reasoning tasks, together with a rigorous curation pipeline with expert validation. **(3)** We provide a comprehensive evaluation of 19 LALMs/OLMs. Our analyses and standardized protocols establish strong baselines and testbeds for future research.

## 2 RELATED WORK

The recent progress of Large Audio-Language Models (LALMs)(Kong et al., 2024; Chu et al., 2024; Wu et al., 2025; Xiaomi, 2025) and Omni-Language Models (OLMs)(Xu et al., 2025; Yao et al., 2024; AI et al., 2025) has significantly advanced audio understanding. At the same time, it has spurred the development of numerous benchmarks to comprehensively evaluate their capabilities. Earlier benchmarks(Wang et al., 2024; Yang et al., 2024) mainly focused on semantic-level understanding tasks (transcription, captioning, and simple question answering), and recent benchmarks(Sakshi et al., 2025; Ma et al., 2025; Kumar et al., 2025) have begun to investigate logical audio reasoning tasks.

However, existing benchmarks largely overlook audio 4D intelligence. Although some advanced benchmarks do touch upon spatio-temporal aspects, their coverage remains limited in both scale and depth. While MMAU Sakshi et al. (2025), MMAU-Pro Kumar et al. (2025) and MMAR Ma et al. (2025) contain temporal questions, they mainly involve identifying the timing or ordering of events (e.g., when a sound occurs, which event comes first). These are primarily perceptual-layer tasks. By contrast, our "temporal deep reasoning" tasks require understanding physical principles or causal dynamics across segments (e.g., inferring how a process evolves over time or how one event implies another), which cannot be solved by local timing cues alone. In addition, the spatial tasks in MMAR and MMAU-Pro are often restricted to single-source localization, and many items do not necessitate meaningful use of stereo cues (e.g., simple arriving vs. departing judgments). In contrast, STAR-BENCH introduces a hierarchical design covering three sub-tasks in complex scenes and explicitly emphasizes stereo-cue-based reasoning.

A comparative overview of STAR-BENCH and prior benchmarks is presented in Tab. 1. STAR-BENCH evaluates deep spatio-temporal reasoning through tasks that go beyond surface-level perception and instead require applying physical or causal knowledge, performing multi-step reasoning in complex real-world scenarios, and integrating information across multiple clips or events. STAR-BENCH rests on a hierarchical and comprehensive task design. In addition, a rigorous data curation pipeline ensures high-quality samples, and robust evaluation strengthens the reliability.

## 3 STAR-BENCH

Understanding dynamic sound sources in both time (1D) and three-dimensional space (3D) is a crucial skill for MLLMs to comprehend the physical world. To address this need, our benchmark, STAR-BENCH, is designed to comprehensively evaluate this 4D intelligence in the audio domain. As illustrated in Fig. 2, our evaluation has two complementary sub-tasks: (1) Foundational Acoustic Perception (Sec. 3.1), which uses procedurally synthesized audio to quantitatively profile a model's

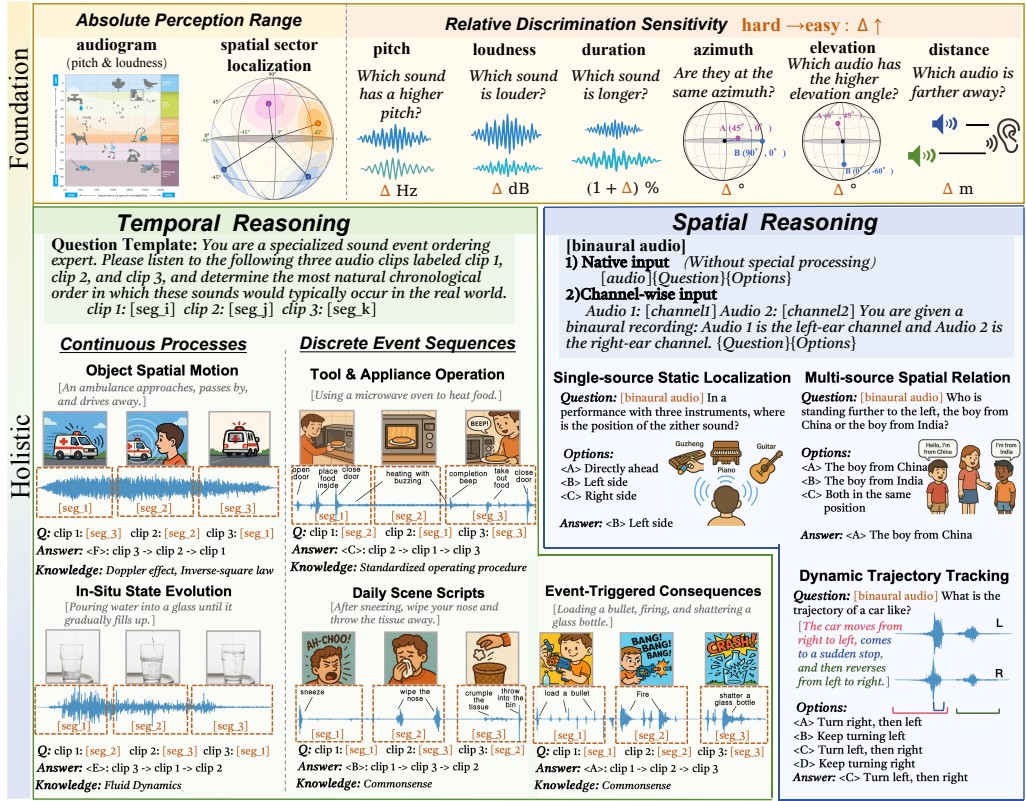

Figure 2: **Data examples from STAR-BENCH**: **(1)** the foundational perception task (upper) and **(2)** the holistic spatio-temporal reasoning task, which includes both temporal reasoning (bottom left) and spatial reasoning (bottom right). Zoom in for the best view.

basic perceptual abilities under controlled conditions, and (2) Holistic Spatio-Temporal Reasoning (Sec. 3.2), which uses real-world audio to evaluate more complex reasoning in dynamic and authentic scenarios. We also elaborate our data curation pipeline in the Sec. 3.3.

## 3.1 FOUNDATIONAL ACOUSTIC PERCEPTION

The Foundational Acoustic Perception task is motivated by the need for a robust, quantitative evaluation of the core perceptual abilities that underpin 4D audio intelligence. A model's capacity for complex reasoning about dynamic audio scenes in the physical world is directly dependent on its ability to accurately perceive fundamental acoustic properties. Our foundational acoustic perception task systematically probes a model's understanding of three critical auditory attributes: **Loudness**, **Pitch**, **Duration**, and the three spatial dimensions: **Azimuth**, **Elevation**, and **Distance**. Just as a solid understanding of grammar is required for writing a complex narrative, a model must be able to accurately perceive these core attributes before it can reason about the dynamic, spatial relationships of sound sources in the physical world. Without a firm grasp of these foundational elements, a model cannot accurately interpret complex, real-world acoustic scenes, which require understanding how sounds change over time and move through space.

We employ a targeted synthesis strategy to generate precise evaluation samples in a controlled environment for the foundational perception task. For non-spatial attributes (Loudness, Pitch, Duration), we synthesize pure sine waves by directly specifying their parameters. For spatial attributes (Azimuth, Elevation, Distance), we use the Pyroomacoustics (Scheibler et al., 2018) physics-based simulation engine to render acoustic scenes. The targeted synthesis strategy allows us to investigate a model's audio perceptual abilities under the following two sub-tasks:

**1) Absolute Perception Range,** which defines the sensory limits of MLLMs for acoustic attributes. For pitch and loudness, we adapt the design of human audiometry tests to create an "audiogram" for the MLLMs. Specifically, we synthesize sine waves with frequencies ranging from 125 Hz to 8000 Hz and loudness levels from $-10$ to 110 dB HL and require the model to identify if a clear beep is

in the first or second part of an audio clip, or if it's not there at all. For spatial attributes, we design interval localization tasks that require the model to identify a sound's azimuth within one of four 90° quadrants (from 0° to 360°), its elevation relative to ear-level (above, at, or below, from -90° to 90°), and its distance category (near, medium, or far, within a 0 - 10m range). Tab. 3 presents detailed examples of these absolute perception range tasks. Through these precise tasks, we establish the absolute limits of what the model can hear, which is crucial for developing AI systems that can safely and effectively interact with the physical world.

**2) Relative Discrimination Sensitivity,** which investigates how well a model can detect small changes in acoustic attributes. The ability to detect small changes allows a model to make nuanced judgments, like determining if a sound is getting louder or a pitch is rising. Analogous to measuring the human Just Noticeable Difference (JND), the relative discrimination task presents the model with an audio clip containing two sounds and requires it to compare them based on a specific attribute. We meticulously designed four to six distinct difficulty levels for each of the six attributes, as detailed in Tab. 3. Level 1 serves as a control group to test for random guessing, presenting identical sounds ($\Delta=0$) for non-spatial attributes and a sub-threshold difference for spatial ones. Subsequent levels then introduce progressively larger differences, ranging from subtle variations perceptible to humans to more significant, real-world changes. By analyzing the model's performance across these different levels of stimulus differences, we can quantitatively assess its discrimination sensitivity for each attribute.

## 3.2 HOLISTIC SPATIO-TEMPORAL REASONING

Building on the model's fundamental audio perceptual abilities (Sec. 3.1), we further introduce holistic temporal reasoning (Sec. 3.2.1) and spatial reasoning (Sec. 3.2.2), which are designed to systematically evaluate a model's reasoning ability that is required for audio 4D intelligence.

### 3.2.1 TEMPORAL REASONING TASKS

The core of temporal reasoning lies in understanding the intrinsic logic of event sequences, encompassing physical causality, functional procedures, or social conventions. To evaluate this capability, we design a novel **Audio Segment Reordering** setting. Specifically, we curate a collection of audio events characterized by strong sequential uniqueness, semantic clarity, and logical universality. Each event is segmented into three clips, which are then shuffled as inputs to the model. The models are required to restore the original temporal sequence based solely on the audio content. Our temporal reasoning tasks are organized into two meta-categories (continuous processes, discrete event sequences) and five subcategories based on their core logical principles.

The **continuous processes** assess a model's ability to track the subtle, continuous evolution of acoustic features within a single, uninterrupted acoustic event. The **object spatial motion** subcategory reconstructs the spatio-temporal trajectory of moving sources (e.g., passing cars, airplanes) by interpreting key acoustic cues, such as the Doppler effect (frequency shifts indicating relative velocity) and the inverse-square law (loudness changes indicating distance). Besides, the **in-situ state evolution** subcategory assesses a model's ability to track the intrinsic evolution of a stationary object's state, a process governed by predictable trend patterns. These trend patterns arise from various underlying principles, including: *Fluid & Pneumatic Dynamics*, where the sound is governed by principles of turbulence, resonance, and pressure changes (e.g., a toilet flushing, water being poured); *Thermodynamic Processes*, involving irreversible state changes driven by heat (e.g., water boiling, food frying); *Energy Decay*, a process governed by resonant decay and frictional damping after a single excitation (e.g., a bell's chime, an explosion's echo); and complex *Biological Rhythms* that reflect an evolving physiological or emotional state.

The **discrete event sequences** category requires the model to understand the logical and temporal relationships between multiple, distinct acoustic events, which are governed by function, convention, or causality. The **tool & appliance operation** sub-category follows the standardized operating procedure for tools and appliances (e.g., a microwave, a power drill), where the sequence is correct when it follows the tool's designed function. The **daily scene scripts** sub-category applies commonsense and contextual script knowledge to follow the conventional sequence of actions in a daily activity (e.g., brushing teeth, drinking water). The **event-triggered consequences** sub-category ap-

plies causal reasoning to infer that a trigger event (e.g., a firework explosion) will be followed by an automatic and irreversible outcome, whether physical (glass shattering) or social (a crowd cheering).

### 3.2.2 SPATIAL REASONING TASKS

Humans effortlessly perceive complex 3D auditory scenes (e.g., hearing a voice from behind, following an approaching car, or locating multiple speakers). Such an ability is fundamental for egocentric interaction and embodied AI systems, for instance, robots that navigate and interact with their surroundings. However, existing benchmarks focus primarily on the localization of static sound sources, whereas real-world scenarios demand reasoning that integrates both spatial and temporal cues. To address this gap, we organize the spatial reasoning task into three subcategories.

The **single-source static localization** evaluates the model's ability to identify the direction of a target sound source among multiple static sources (e.g., judging whether a sound comes from the left or right). It assesses the basic spatial perception capability of the model and provides the foundation for more advanced reasoning. The **multi-source spatial relation** requires the model to determine the relative spatial relationships among multiple simultaneous sound sources (e.g., comparing the placement of two speakers to decide which one is further to the right). Beyond localizing each source individually, the model must infer their spatial placement and choose the appropriate relational description from multiple candidates. The **dynamic trajectory tracking** introduces moving sound sources, which require the model to go beyond basic spatial perception to dynamically model spatio-temporal relations for reasoning about complex movement trajectories (e.g., tracking a passing car moving from left to right). This task extends spatial reasoning into the temporal domain and is more faithful to the complexity of real-world acoustic scenarios.

However, evaluating existing LALMs on multi-channel spatial tasks is challenging. The common practice of these models is to average multi-channel audio into a mono signal, resulting in the loss of substantial spatial information. We conduct a simple experiment as shown in Fig. 3. We construct 20 pseudo-stereo signals by assigning the original audio to the left channel and its additive inverse to the right. While human listeners could easily perform sound event classification on these signals, the models consistently failed due to signal cancellation during the mono conversion. The

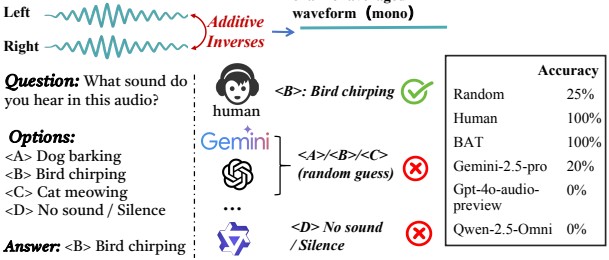

Figure 3: Audio preprocessing in existing models results in the loss of dual-channel information.

result confirms their lack of explicit support for genuine stereo audio processing. To provide a comprehensive assessment, we adopt two complementary strategies. The first is the **native input** setting, where the model directly processes stereo audio using its default pipeline. This allows us to probe its intrinsic ability to exploit spatial cues. The second is the **channel-wise input** setting, where the left and right channels are presented separately with explicit textual instructions, as shown in the bottom right of Fig. 2. This configuration serves as an ablation study to examine whether current models have any spatial capability when the binaural information is preserved at the input.

### 3.3 DATA CURATION PIPELINE

Our data curation pipeline integrates procedural synthesis with real-world data collection to ensure both comprehensive coverage and ecological validity. Fig. 4 shows the distribution and statistics of our STAR-BENCH. All audio for the *foundational perception* task is synthesized using precise parameterization or the Pyroomacoustics (Scheibler et al., 2018) physics-based simulator, providing complete control over acoustic parameters. Domain experts rigorously validate the task difficulty levels, which are then calibrated through human testing. For the *holistic spatio-temporal reasoning* task, the curation process comprises four key stages (see Fig. 5):

**1)** Taxonomy Construction and Data Sourcing: We build a hierarchical task taxonomy through a collaborative process involving domain experts and the Gemini 2.5 Pro (Comanici et al., 2025). This framework guides the sourcing of candidate data from large-scale, real-world audio libraries: Clotho

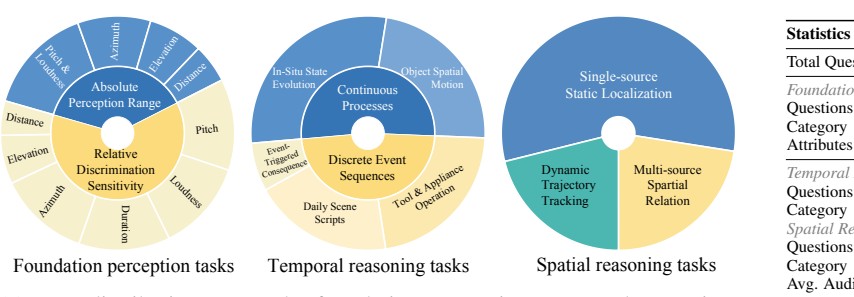

(a) Data distribution across the foundation perception, temporal reasoning, and spatial reasoning three tasks.

| Statistics | Number |
|---|---|
| Total Questions | 2,353 |
| *Foundation Perception* | |
| Questions | 951 |
| Category | 2 |
| Attributes | 6 |
| *Temporal Reasoning* | |
| Questions | 900 |
| Category | 2 |
| *Spatial Reasoning* | |
| Questions | 502 |
| Category | 3 |
| Avg. Audio Length | 14.03 sec |

(b) Statistics.

Figure 4: **(a)** The **data distribution** of STAR-BENCH across three main tasks. **(b)** **Data statistics** of our benchmark, including the total number of questions for each task and their sub-categories, and the average audio length for reasoning tasks.

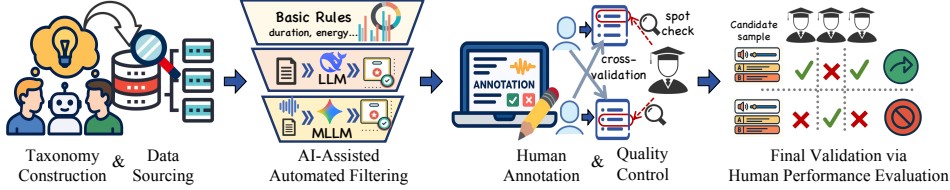

Figure 5: The four-stage **data annotation pipeline** for constructing our STAR-BENCH.

(Drossos et al., 2019) and FSD50K (Fonseca et al., 2022) for temporal reasoning, and STARSS23 (Shimada et al., 2023), along with audio sourced from the internet for spatial reasoning.

**2)** AI-Assisted Automated Filtering: This process employs an efficient three-stage funnel. First, we discard unsuitable samples based on basic properties like duration and energy. Next, an LLM (e.g., DeepSeek-V3 (Liu et al., 2024a)) performs an initial screening based on textual metadata, providing justifications for its decisions. Finally, a powerful multimodal model (e.g., Gemini 2.5 Pro (Comanici et al., 2025)) analyzes the audio, metadata, and the LLM's outputs. The final step yields a judgment, a quality score, and a preliminary classification, further filtering irrelevant samples. The detailed prompts used to query the LLMs are provided in Appendix B.3.1.

**3)** Human Annotation and Quality Control: We recruit and train 10 undergraduate annotators to label the data using a professional platform. During this process, AI-generated information is provided as an auxiliary reference. To ensure high-quality labels, we implement a stringent two-round review process: the first round involves inter-annotator cross-validation until a consensus is reached, while the second consists of random spot-checks by three domain experts. More details are provided in Appendix B.3.2.

**4)** Final Validation via Human Performance Evaluation: To ensure all items in the benchmark are fair, unambiguous, and solvable by humans, we implement a final validation stage. In this phase, domain experts act as examinees and solve our tasks. Only items that are independently and correctly solved by at least two-thirds of the experts are retained. Our rigorous protocol ensures that all problems in our benchmark are well-posed and reliably solvable by human experts.

## 4 EVALUATION

**Benchmarking Models.** Our evaluation covers 19 models (16 open-source and 3 closed-source models). The open-source models span three categories: (1) Large Audio Language Models designed for universal audio-text understanding, including SALMONN (Tang et al., 2024), Qwen2-Audio Instruct (Chu et al., 2024), Audio Flamingo 3 (Goel et al., 2025) with its 'think' variant, DeSTA2.5-Audio (Lu et al., 2025), Kimi-Audio (KimiTeam et al., 2025), Step-Audio-2-mini (Wu et al., 2025), MidashengLM (Dinkel et al., 2025), and Xiaomi-MiMo-Audio (Xiaomi, 2025) with its 'think' variant; (2) a specialized model for spatial audio, BAT (Zheng et al., 2024); and (3) Omni Language Models with fully multimodal support, including Qwen-2.5-Omni (Xu et al., 2025), Phi4-MM (Abouelenin et al., 2025), Gemma-3n-E4B-it (Team et al., 2025), and Ming-Lite-Omni-1.5 (AI et al., 2025). We also include three leading closed-source models: Gemini 2.5 Pro (Comanici

Table 2: Evaluation results of various models on STAR-BENCH. The best performance is highlighted in **bold**, and the second-best ones are underlined. MA (Macro Accuracy) denotes the unweighted mean of class-wise accuracies, while OA (Overall Accuracy) denotes the proportion of correctly answered instances. All reported values are AA (Average Accuracy across multiple runs) only; for ACR (All-Correct Rate), see Appendix D.

| Models | Size | Foundational Perception | | | Temporal Reasoning | | | Spatial Reasoning | | | | MA (%) |
|---|---|---|---|---|---|---|---|---|---|---|---|---|
| | | Range | Sensitivity | MA | Continuous | Discrete | OA | Localization | Relation | Trajectory | OA | |
| Random Guess | - | 23.75 | 26.38 | 25.33 | 14.29 | 14.29 | 14.29 | 33.33 | 33.33 | 33.33 | 33.33 | 24.32 |
| Human | - | 79.42 | 74.55 | 75.60 | 90.12 | 85.51 | 88.00 | 70.00 | 80.00 | 77.00 | 73.72 | 79.11 |
| SALMONN [33] | 13B | 27.32 | 25.48 | 26.22 | 14.88 | 13.30 | 14.15 | 26.15 | 28.61 | 39.94 | 29.62 | 23.33 |
| Audio Flamingo 3 [14] | 8.4B | 31.79 | 35.72 | 34.15 | 9.23 | 8.01 | 8.67 | 37.22 | 38.35 | 44.03 | 38.91 | 27.24 |
| Audio Flamingo 3 think [14] | 8.4B | 25.54 | 34.08 | 30.66 | 13.22 | 14.02 | 13.59 | 35.45 | 37.46 | 38.05 | 36.45 | 26.90 |
| Qwen2-Audio-Instruct [7] | 8.4B | 29.88 | 26.47 | 27.84 | 13.29 | 12.10 | 12.74 | 21.32 | 24.78 | 15.09 | 20.78 | 20.45 |
| DeSTA2.5-Audio [26] | 8.8B | 29.87 | 19.79 | 23.82 | 16.53 | 17.39 | 16.93 | 23.67 | 34.81 | 37.74 | 29.15 | 23.30 |
| BAT [51] | 7B | 22.81 | 6.25 | 12.87 | 0.00 | 0.00 | 0.00 | 0.00 | 0.00 | 0.00 | 0.00 | 4.29 |
| Phi4-MM [1] | 5.5B | 19.14 | 29.85 | 25.56 | 16.74 | 16.99 | 16.85 | 33.10 | 27.14 | 34.28 | 32.01 | 24.81 |
| Kimi-Audio [18] | 7B | 23.29 | 27.50 | 25.82 | 19.97 | 16.83 | 18.52 | 27.56 | 38.94 | 44.03 | 33.60 | 25.98 |
| MiDashengLM [10] | 7B | 36.94 | 30.78 | 33.24 | 15.43 | 17.31 | 16.30 | 43.11 | 45.43 | 46.23 | 44.29 | 31.28 |
| Step-Audio-2-mini [39] | 7B | 29.65 | 27.14 | 28.14 | 15.36 | 15.87 | 15.59 | 33.33 | 31.27 | 37.74 | 33.80 | 25.84 |
| Gemma-3n-E4B-it [34] | 7.5B | 18.55 | 25.02 | 22.43 | 16.87 | 16.27 | 16.59 | 23.32 | 41.89 | 33.96 | 29.75 | 22.92 |
| Ming-Lite-Omni-1.5 [3] | 18.9B | 26.76 | 26.76 | 26.76 | 17.08 | 15.54 | 16.37 | 20.14 | 35.10 | 38.36 | 27.35 | 23.49 |
| Qwen-2.5-Omni [43] | 7B | 28.76 | 32.32 | 30.90 | 16.32 | 17.71 | 16.96 | 39.46 | 41.30 | 27.04 | 37.25 | 28.37 |
| Xiaomi-MiMo-Audio [40] | 7B | 34.95 | 31.59 | 32.93 | 18.18 | 19.15 | 18.63 | 36.16 | 41.30 | 45.28 | 39.24 | 30.27 |
| Xiaomi-MiMo-Audio-think [40] | 7B | 29.90 | 24.93 | 26.92 | 16.80 | 19.39 | 18.00 | 34.28 | 44.54 | 36.79 | 37.12 | 27.35 |
| MiniCPM-O-v2.6 [48] | 8B | 31.02 | 31.87 | 31.53 | 15.36 | 17.39 | 16.30 | 29.92 | 43.36 | 38.36 | 34.73 | 27.52 |
| GPT-4o Audio [2] | - | 27.58 | 34.55 | 31.76 | 15.91 | 23.56 | 19.44 | 41.81 | 43.97 | 39.94 | 41.70 | 30.97 |
| Gemini 2.5 Flash [8] | - | 33.46 | 43.88 | 39.72 | 27.55 | 34.38 | 30.70 | 24.62 | 43.07 | 22.64 | 28.35 | 32.92 |
| Gemini 2.5 Pro [8] | - | 39.90 | 51.13 | 46.64 | 54.88 | 62.74 | 58.52 | 40.87 | 48.97 | 45.28 | 43.62 | 49.59 |

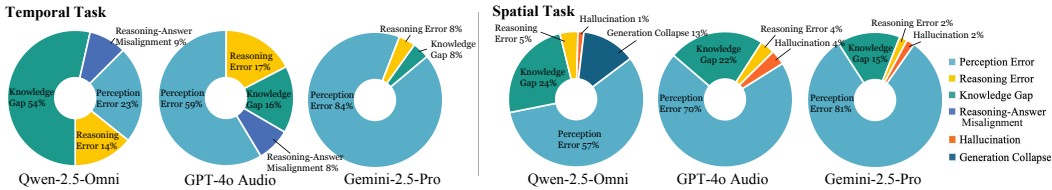

Figure 6: Error distribution across temporal and spatial Tasks.

et al., 2025) (updated June 2025), Gemini 2.5 Flash (updated June 2025), and GPT-4o-audio-preview (Achiam et al., 2023) (version 2025-06-03).

**Robust Evaluation.** All questions in STAR-BENCH are presented as multiple-choice questions and evaluated using classification accuracy, with correctness determined via string matching of option labels or their full text. To ensure robustness, we evaluate each question multiple times under minor prompt perturbations, a strategy detailed in Appendix C. This approach yields two key metrics: **Average Accuracy (AA)**, the mean accuracy across all runs, and **All-Correct Rate (ACR)**, the proportion of questions answered correctly in every run, which serves as a stronger indicator of model reliability. Due to space limitations, we primarily report AA in the main text, while complete experimental results are available in Appendix D.

## 4.1 MAIN RESULT ANALYSIS

We present a comprehensive evaluation on STAR-BENCH, as shown in Tab. 2. Due to the space limit, detailed results on each task are provided in Appendix D. Our key findings are as follows:

**STAR-BENCH is Challenging** STAR-BENCH presents a considerable challenge for existing models. Human evaluators achieve high accuracy across all task categories (e.g., 75.6% on perception, 88.0% on temporal, and 73.7% on spatial tasks), whereas all tested models fall well below this baseline. Most open-source models perform close to random guessing, and even the best closed-source model, Gemini 2.5 Pro, reaches only 49.59% average accuracy. In addition, model predictions on STAR-BENCH exhibit low reliability, as evidenced by the pronounced gap between their Average Accuracy (AA) and All-Correct-Rate (ACR) scores. A detailed discussion of this issue is provided in Appendix E.1. Although the underlying audio data for the temporal tasks (e.g., FSD50K, Clotho) is commonly used for model pre-training, our novel task formulation of temporal reasoning deliberately departs from conventional audio QA formats. This design allows for a more thorough evaluation of the integrated capabilities of current models. Meanwhile, this design also serves as a diagnostic lens on the limitations of current training pipelines. The poor performance across models suggests that existing training paradigms often centered on clip-level tagging, QA, or captioning

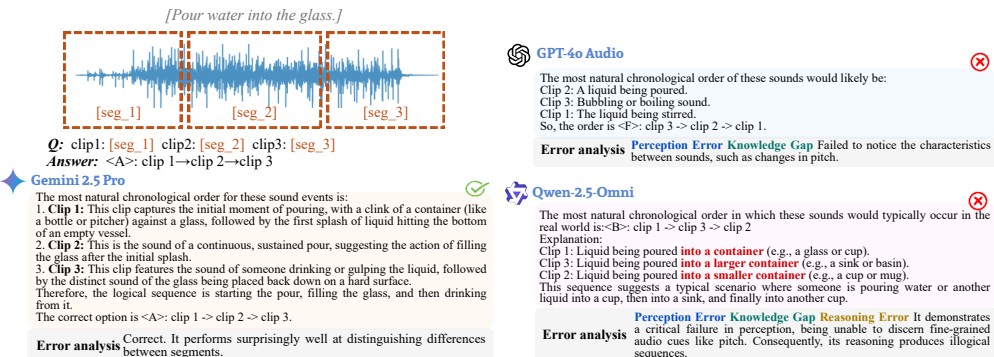

Figure 7: An error case in temporal reasoning task. More cases are provided in the Appendix F.

over linguistically salient cues (e.g., using FSD50K for sound event recognition) and do not equip models with the abilities needed for audio 4D intelligence.

**A Clear Performance Gap between Closed-Source and Open-Source Models** On the foundational perception and temporal tasks, Gemini 2.5 Pro establishes a commanding lead among all models. On spatial tasks, however, nearly all models, both closed- and open-source, perform poorly. As indicated by the prior experiment (Fig. 3), this is likely because most models (except BAT) discard multi-channel information during preprocessing, thereby losing key acoustic cues needed for spatial reasoning. Among closed-source models, Gemini 2.5 Pro surpasses Gemini 2.5 Flash, suggesting that stronger reasoning capabilities deliver substantial gains. In contrast, open-source models show the opposite pattern: the "think" modes of Audio Flamingo 3 and Xiaomi-MiMo-Audio perform worse than their no-thinking counterparts, implying that without sufficiently solid perceptual and knowledge foundations, reasoning can be ineffective or even detrimental.

## 4.2 DISCUSSION: WHY DO EXISTING MODELS STRUGGLE ON STAR-BENCH?

To better understand the underlying causes of the poor performance of existing models, we conduct a detailed error analysis along with a series of ablation studies. Due to space limitation, the ablation study on spatial reasoning is provided in Appendix E.2.

**Error Analysis.** We conduct a manual error analysis on 200 failed predictions sampled equally from temporal and spatial tasks of three representative models (Gemini 2.5 Pro, GPT-4o-audio, and Qwen-2.5-Omni). For temporal tasks, our analysis reveals a clear capability hierarchy across the models. The open-source Qwen-2.5-Omni shows major deficiencies in all three core abilities: its perception is coarse-grained and unable to capture subtle inter-segment distinctions, and a substantial knowledge gap (54%) leads to reasoning that often appears specious due to the absence of physical-world grounding. GPT-4o-audio demonstrates stronger knowledge, but still suffers from perceptual and reasoning limitations, along with low-level issues such as misalignment between reasoning and final answers. In contrast, Gemini 2.5 Pro excels in knowledge and reasoning, shifting its primary bottleneck to the more advanced challenge of fine-grained perception (84%). As shown in Fig. 7, Gemini 2.5 Pro is the only model to succeed by providing a remarkably detailed description of acoustic nuances. Our finding suggests that the **advanced world knowledge is deeply embedded within detailed audio-text captioning.** While open-source models largely remain at a coarse semantic level (e.g., sound event classification), our analysis highlights that enabling them to generate fine-grained acoustic descriptions is critical toward more robust reasoning. On the other hand, most models demonstrate a lack of native spatial awareness in audio tasks, with weaknesses in perception, knowledge, and reasoning. Additionally, a prevalent type of error involves vision-centric hallucinations (e.g., "...based on the car's trajectory in the video..."). This may be attributable to the models' training on visual spatial tasks, leading them to misapply visual reasoning to auditory inputs.

**Lack of Human-like Range and Sensitivity in Foundational Perception.** To quantify the gap in perceptual range and sensitivity, we provide detailed visualizations of model performance on our foundational perception tasks in Fig. 8. The first row of Fig. 8 presents audiograms that compare model coverage across the pitch–loudness space. Gemini 2.5 Pro achieves a much broader coverage than the other two models, where greener regions indicate higher accuracy and the covered area reflects the perceptual range. In contrast, human listeners with normal hearing are expected to

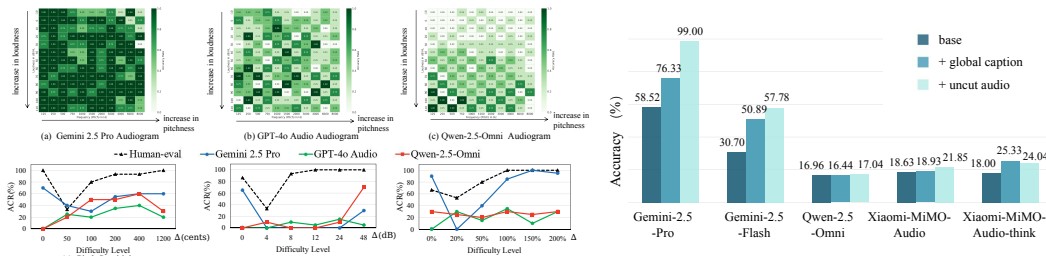

Figure 8: The **range** and **sensitivity analysis** in foundational perception.

Figure 9: The **ablation study** on temporal reasoning.

achieve near-full coverage, underscoring the gap between current models and human perceptual abilities in terms of range. The second row of Fig. 8 further track the performance of both models and human subjects on the three core acoustic attributes (pitch, loudness, and duration) as task difficulty decreases. The results reveal a stark performance gap between all models and the human baseline, particularly in the perception of fine-grained loudness differences. A clear trend is visible even for the top-performing Gemini 2.5 Pro: its accuracy, while competent on easier tasks, plummets as perceptual granularity increases. This directly corroborates our error analysis, identifying fine-grained perception as its primary bottleneck. Notably, its performance on duration perception is an exception, showcasing **temporal grounding capabilities superior to those of other models** by accurately assessing audio segment lengths.

**Ablation Study on Temporal Reasoning.** To further pinpoint the specific limitations of temporal reasoning, we augment the baseline audio segment reordering task with two progressively easier settings: (1) + *Global Caption*, where a single sentence describing the overall scene is provided as a contextual guide; and (2) + *Uncut Audio*, where the complete, unsegmented audio track is offered as a reference, reducing the task to a straightforward process where the correct order can be determined simply by comparing and grounding each segment within the full audio. As shown in Fig. 9, Gemini 2.5 Pro's performance scales effectively with task simplification, culminating in a near-perfect 99% accuracy in the + *Uncut Audio* setting. In contrast, the open-source models show minimal to no improvement across these settings. Their performance remains stagnant even when provided with the complete audio reference, despite the simplified nature of the task. This finding starkly exposes a core weakness in current open-source models: **a fundamental inability to effectively compare, ground, and integrate information from multiple audio inputs.**

## 5 CONCLUSION

We introduce STAR-BENCH, a comprehensive benchmark for evaluating 4D audio intelligence over time and 3D space. We use rigorous human annotation, consensus review, and expert validation to ensure the high quality of data samples. STAR-BENCH establishes standardized tasks and protocols for studying 4D audio intelligence, offering actionable diagnostics for model developers. We expect STAR-Bench to accelerate progress on advanced audio models and training with spatialized corpora, capabilities that are crucial for embodied agents.

## ACKNOWLEDGMENTS

This project is funded in part by Shanghai Artificial Intelligence Laboratory, Shanghai Innovation Institute, the Centre for Perceptual and Interactive Intelligence (CPII) Ltd under the Innovation and Technology Commission (ITC)'s InnoHK. Dahua Lin is a PI of CPII under the InnoHK.

ETHICS STATEMENT

Our study primarily relies on datasets from open-source research communities and publicly available online resources, as described in detail in the main text. These datasets do not involve private information, sensitive content, or material that could raise concerns related to safety, discrimination, or harmful societal impact. All annotation and evaluation tasks were carried out by university volunteers who participated on a voluntary basis. No human subjects were placed at risk, and no personally identifiable information was collected during the course of this research.

REPRODUCIBILITY STATEMENT

We provide a detailed description of the construction process of our benchmark dataset and evaluation pipeline in the main text. To facilitate reproducibility, we will release the benchmark dataset as well as the evaluation code to the community. Clear documentation and step-by-step instructions are included to ensure that other researchers can replicate our experiments and verify the reported results.

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

## THE USE OF LARGE LANGUAGE MODELS

We used Gemini-2.5-Pro to assist in expanding and consolidating the taxonomy of tasks in our benchmark. Both DeepSeek-V3 and Gemini-2.5-Pro were utilized for the automated pre-screening of candidate data. The final task definitions and data samples are verified by humans. We also used GPT-4o to generate some of the illustrative figures presented in the paper, and used GPT-5 to polish the manuscript text. Only human-verified revisions are included in the final version.

## A   RELATED WORK

### A.1   AUDIO LANGUAGE MODELS

With the advancements of large language models (LLMs) and multimodal language models (Yang et al., 2025a; Jiang et al., 2024; Achiam et al., 2023; Comanici et al., 2025; Cai et al., 2024; Touvron et al., 2023; Liu et al., 2024c; 2025; Zhang et al., 2025b; Qi et al., 2025; Xing et al., 2025b;a; Ding et al., 2025; Wei et al., 2025a;b; Li et al., 2025; Zhang et al., 2025a), recent research has increasingly focused on integrating audio perception with LLMs to enhance audio understanding and reasoning. Existing methods can be broadly grouped into two categories: Large Audio Language Models(LALMs) and Omni Language Models(OLMs).

Most LALMs combine a pre-trained audio encoder with an LLM backbone, where the two modalities are aligned via large-scale text-audio joint training. Notable models include LTU-AS (Gong et al., 2023), SALMONN (Tang et al., 2024), MidashengLM (Dinkel et al., 2025), Audio Flamingo series (Ghosh et al., 2025; Goel et al., 2025), Qwen-Audio series (Chu et al., 2023; 2024), Step-Audio (Wu et al., 2025) and Mimo-Audio (Xiaomi, 2025). These models have achieved remarkable performance across a wide range of audio understanding tasks, including automatic speech recognition(ASR), spoken question answering(SpokenQA), and automated audio captioning(AAC). In parallel, OLMs extend this paradigm to unify multimodal understanding with representative examples such as Qwen-2.5-Omni (Xu et al., 2025), Ming-Omni (AI et al., 2025),MiniCPM-O (Yao et al., 2024), Phi-4 (Abouelenin et al., 2025), GPT-4o (Achiam et al., 2023), and Gemini 2.5 (Comanici et al., 2025). Notably, they also achieve impressive performance on audio understanding and reasoning, highlighting their potential to bridge multimodal perception and advanced audio intelligence.

### A.2   AUDIO BENCHMARKS

Existing audio benchmarks illustrate the rapid progress of multimodal evaluation but also expose limitations. AudioBench (Wang et al., 2024) and AIR-Bench (Yang et al., 2024) primarily focus on tasks such as automatic speech recognition (ASR), spoken question answering (SpokenQA), and audio captioning (AAC). These settings tend to reduce audio understanding to transcription or description, thereby neglecting the broader spectrum of acoustic reasoning. MMAU (Sakshi et al., 2025) and MMAR (Ma et al., 2025) further extend the evaluation scope. However, their results reveal an inherent weakness—LLMs equipped with audio captions can perform on par with advanced LALMs, suggesting that such benchmarks still probe little beyond language-level semantics.

Although some advanced benchmarks, such as MMAR (Ma et al., 2025) and MMAU-Pro (Kumar et al., 2025), do touch upon spatio-temporal aspects, their coverage remains limited in both scale and depth. For instance, their temporal analysis is typically reduced to identifying the timing or order of events occurring in the audio, while spatial analysis is often limited to localizing a single sound source. In contrast, our benchmark systematically evaluates models' temporal and spatial deep reasoning capabilities within complex, real-world physical contexts, requiring them to infer causal and dynamic relationships. Beyond audio benchmarks, multimodal benchmarks in video question answering (Cheng et al., 2025; Yang et al., 2025c) and embodied AI (Yang et al., 2025b) have emphasized temporal and spatial reasoning. However, these frameworks are predominantly grounded in the visual modality, where exploration of the audio modality remains comparatively limited. In real-world scenarios, audio understanding often depends on integrating information across multiple sound streams and reasoning about subtle changes in intensity, phase, or frequency—capabilities that existing benchmarks scarcely capture.

| Attribute | Range / Level | Example |
|---|---|---|
| **Absolute Perception Range** | | |
| Pitch, Loudness | 125 Hz - 8000 Hz
-10dB - 110dB | *[Audio]The audio you just heard is divided into two halves.*
*Does a sound appear in the first half, the second half, or is it not present at all?*
(A) The first half (B) The second half (C) It is not present at all (D) Unable to determine |
| Azimuth | 0° - 360° | *[Audio] Given that 0° is directly in front and the angle increases clockwise, which azimuth range is the sound most likely coming from?*
(A) Front-Right (0°–90°) (B) Back-Right (90°–180°) (C) Back-Left (180°–270°) (D) Front-Left (270°–360°) (E) Unable to determine |
| Elevation | -90° - 90° | *[Audio] Where does the sound seem to be coming from in terms of elevation, relative to ear level?*
(A) Above ear level (B) Below ear level (C) At ear level (D) Unable to determine |
| Distance | 0 meter - 10 meters | *[Audio] How far away does the sound seem to be?*
(A) Near (within about 0–3 meters) (B) Medium (around 3–8 meters) (C) Far (more than 8 meters) (D) Unable to determine |
| **Relative Discrimination Sensitivity** | | |
| Pitch | 0, 50, 100, 200, 400, 1200 (cents) | *[Audio] Which sound has a higher pitch: the first sound, the second sound, or are they the same?*
(A) The first sound has a higher pitch (B) The second sound has a higher pitch(C) Both sounds are the same (D) Unable to determine |
| Loudness | 0, 4, 8, 12, 24, 48 (dB) | *[Audio] Which sound is louder: the first sound, the second sound, or are they the same?*
(A) The first sound is louder (B) The second sound is louder (C) Both sounds are the same (D) Unable to determine |
| Duration | 0, 20, 50, 100, 150, 200 (%) | *[Audio] Which sound is longer: the first sound, the second sound, or are they the same?*
(A) The first sound is longer (B) The second sound is longer (C) Both sounds are the same (D) Unable to determine |
| Azimuth | 30, 60, 90, 120, 150, 180 (°) | *Audio 1: [Audio_1] Audio 2:[Audio_2] Are Audio 1 and Audio 2 at the same azimuth? (Consider differences of less than 45° as the same.)*
(A) Same (B) Different (C) Unable to determine |
| Elevation | 15, 90, 120, 150 (°) | *Audio 1: [Audio_1] Audio 2:[Audio_2] Which audio has the higher elevation angle? (Consider differences of less than 45° as the same.)*
(A) Audio 1 is higher (B) Audio 2 is higher (C) Both are at the same elevation (D) Unable to determine |
| Distance | 1-2, 4-5, 6-7, 8-9 (meters) | *Audio 1: [Audio_1] Audio 2:[Audio_2] Which audio is farther away? (Consider differences of less than 3 meters as the same.)*
(A) Audio 1 is farther away (B) Audio 2 is farther away (C) Both audios are the same (D) Unable to determine |

Table 3: Task examples of foundational acoustic perception.

Our benchmark aims to address these gaps by introducing tasks that require **multi-audio input and cross-audio reasoning**, such as comparing or integrating information across multiple sound inputs, as well as **fine-grained spatio-temporal deep reasoning**, such as tracking how acoustic patterns evolve with underlying physical changes. Rather than being limited to surface-level semantics, the benchmark is designed to assess whether models can leverage raw audio cues to perform physically grounded reasoning across spatial and temporal dimensions.

## B    Details of Data Annotation

In this section, we present the details of data annotation.

### B.1    Prompts for audio captioning

The prompt for Gemini 2.5 Pro audio captioning: "Please provide a detailed description of the audio, including speech, music, environmental sounds, and any other noticeable elements. Be as specific as possible."

### B.2    Detail information for foundational acoustic perception

Tab. 3 details the ranges and levels used for each acoustic attribute, alongside illustrative examples of our foundational acoustic perception tasks.

#### B.2.1    Binaural Audio Synthesis

We generated binaural recordings for foundational perception tasks (azimuth, elevation, distance) in Pyroomacoustics (Scheibler et al., 2018) across three rectangular rooms—small (4.0×3.5×2.8 m), medium (8.0×6.0×3.5 m), and large (20×15×8 m)—each with a frequency-independent wall absorption coefficient of 0.25. Image-source reflections were modeled up to order 10 at 44.1 kHz (matched to the HRTF sampling rate). For each room, we evaluated two listener positions (distinct Cartesian coordinates) and oriented the head toward the +x axis. Binaural reception used a co-located two-microphone array at the listener position with ear-specific directivity derived from a measured SOFA HRTF[1] (MIT KEMAR, "normal pinna"; interpolation order 12, 1000 points), loaded via a local SOFA reader and applied to the left/right channels.

For each condition (room × listener), sources were placed on a sphere centered at the listener (radii 1–10 m; configurable azimuth/elevation), and ear-specific BRIRs were computed. Mono source signals were drawn from three curated audio clips ("alarm," "applause," "telephones"), downmixed if necessary. Rendering was performed by convolving each dry signal with the left/right BRIRs after an early/late mix to emphasize distance cues: we preserved the first 80 ms and attenuated the

---

[1] https://sofacoustics.org/data/database/mit/mit_kemar_normal_pinna.sofa

late tail by 0.5. We then applied global peak normalization across the batch to avoid clipping while preserving inter-position level differences.

We discretized each attribute into fixed partitions to control dataset balance.

**Absolute azimuth:** Eight angles $\{30°, 60°, 120°, 150°, 210°, 240°, 300°, 330°\}$. For each angle we rendered all combinations of 3 rooms $\times$ 2 listener positions $\times$ 2 source clips, yielding $8 \times (3 \times 2 \times 2) = 96$ utterances. **Absolute elevation:** Six angles $\{-75°, -45°, -15°, 15°, 45°, 75°\}$. Per angle we rendered 3 rooms $\times$ 2 listener positions $\times$ 2 source clips, for $6 \times (3 \times 2 \times 2) = 72$ utterances. **Absolute distance:** Radii from 1–10 m with a nonuniform allocation to emphasize near-field cues: for 1–7 m we generated 6 utterances per meter (42 total), and for 8–10 m we generated 3 per meter (9 total), giving $42 + 9 = 51$ utterances per (room $\times$ listener) set.

**Relative azimuth:** Differences were multiples of $30°$: $\{30°, 60°, 90°, 120°, 150°, 180°\}$ (6 levels), totaling $6 \times 20 = 120$ utterances. **Relative elevation:** Four difference angles $\{15°, 90°, 120°, 150°\}$ with 18, 17, 17, 12 utterances respectively (64 total). **Relative distance:** Four difference levels $\{1-2, 4-5, 6-7, 8-9\}$ m with counts per level $\{12, 12, 12, 9\}$, totaling 45 utterances.

### B.3 Details of the Curation Process for Reasoning Tasks

### B.3.1 Prompt Used for AI-Assisted Filtering of Temporal Task Data

Fig. 10 and Fig. 11 present our carefully designed prompts, which leverage Gemini 2.5 Pro to filter candidate data that meet the requirements of audio segment reordering. Briefly, we feed the audio, its metadata, and our task description, and ask Gemini 2.5 Pro to decide, under our strict criteria of strong sequence uniqueness, semantic clarity, and high logical universality, (i) whether the audio is suitable for a reordering task, (ii) whether it reflects a continuous or discrete process, (iii) the reasoning behind its judgment, and (iv) a quality score. We adopt a conservative filtering strategy, discarding only samples explicitly marked as "not applicable". All remaining clips, along with the model's analysis, are then passed to professional annotators for verification and annotation. A prior LLM-based filtering step follows a similar procedure, but without audio input.

### B.3.2 Details of Human Annotation and Quality Control

Following automated filtering, each candidate sample undergoes a rigorous, multi-stage human annotation and quality control process to ensure high data quality and annotation consistency. This process is as follows:

(1) **Systematic Training:** All annotators received detailed written guidelines and completed a trial annotation of 10 samples. These trials were meticulously reviewed by experts to ensure a unified understanding of the criteria.

(2) **Inter-annotator Cross-validation:**

   (i) *Initial Annotation:* A sample is first annotated by Annotator A. The annotation content includes:
   - For Temporal Reasoning: Task compliance checks, segment boundary delineation, textual descriptions for sub-clips and the global audio, scene classification, and audio quality scoring.
   - For Spatial Reasoning: Selecting appropriate segments, task classification, and the generation of a question, the correct answer, and distractor options for the multiple-choice format.

   (ii) *Review and Flagging:* The annotated sample is then fully reviewed by Annotator B, who flags any inconsistencies with detailed comments and marks the sample as "failed".

   (iii) *Consensus through Negotiation:* Annotators A and B then discuss all flagged issues to reach a consensus and apply corrections. During the discussions, primary sources of ambiguity are as follows:
   - For Temporal Reasoning: (a) The reasonableness of the segmented clip boundaries. (b) The existence of multiple logically plausible orderings for the segmented clips. (c) Significant discrepancies in audio quality scores.(d) Adherence to formatting and content guidelines for the captions.

- For Spatial Reasoning: (a) Whether the spatial perception presented in the audio unambiguously aligns with the annotated answer. (b) Whether the constructed question-answer pair clearly necessitates the use of audio spatial cues for resolution. (c) Potential ambiguity in mapping the event name mentioned in the question to a specific sound in the audio. (d) The appropriate difficulty and plausibility of the distractor options.

(iv) *Expert Arbitration:* In the cases where a consensus cannot be reached, the sample is escalated to an expert panel for a final decision. If the experts still cannot agree, the sample is discarded.

(3) **Expert Spot-check:** After passing cross-validation, a random 10% of samples undergo a final quality check by experts to ensure consistency and accuracy. Any discovered issues are then sent back for revision.

---

# Role Setting:
You are a rigorous audio analysis expert, specializing in identifying dynamic audio with explicit temporal logic conforming to physical laws or strong causality. Your task is to screen suitable audio samples for high-standard "Audio Sequence Ordering Evaluation." Analysis should rely on the audio itself, with text as auxiliary reference.

# "Audio Clip Ordering" Evaluation Task:
Qualified audio is segmented into three clips, shuffled, and given to the model, which must reconstruct the sequence using only sound. Candidate audio must meet the following strict standards:
1.Strong sequence uniqueness: The events in the audio must present a unique and clearly discernible temporal progression, with no possibility of alternative plausible orderings.
2.Semantic clarity: Events in the audio must be easily identifiable by sound alone.
3.High logical universality: The event sequence should conform to commonsense physical laws or strong causal relations, such that listeners from different backgrounds can reach a consistent understanding.
Note: Since the model sees no text, samples must be interpretable solely from sound.

# Audio Classification Standards:
1. Sortable Single Event:
- Definition: Audio primarily represents a continuous event driven by a single process, exhibiting significant and predictable temporal dynamics.
- Possible categories include (but are not limited to):
  Spatial movement and distance variation. Physical processes in progression. Energy or state decay. Biological activity dynamics.
- Core judgment: The core judgment is whether the change is governed by a single continuous physical process, is significant, and is commonly recognized as unambiguous.
- Examples:
  The sound of pouring water.  A ball bouncing to rest.
2. Sortable Multi-Event:
- Definition: Audio contains two or more independent events, where the events exhibit a strong causal relationship such that "A inevitably leads to B" or "A must precede B."
- Core judgment: The core judgment is whether sub-events have distinct, separable acoustic features, follow a direct and widely recognized causal chain, and together form a concise, complete, and unambiguous process.
- Examples:
  Opening a bottle → pouring water → setting down the bottle.
  Cracking an egg → stirring → pouring into an oil pan.
Note: If event boundaries are unclear but the overall structure forms a continuous process, classify as "Single Event." If there are clearly distinct stages with evident logical links, classify as "Multi-Event."

#Special Exclusion Rules：
Filter out the following audio types, even if they exhibit some "dynamic change." If they fail semantic clarity, sequence uniqueness, or logical universality, they must be labeled "Not Applicable."
- Static or repetitive sounds.
- Sounds with intensity variation but without reflecting a physical law or typical action sequence.
- Weak or absent causality, or dependent on special background knowledge.
- Ambiguous, subjective, or interpretable in multiple ways.
- Ambient noise or atmosphere shifts without structured events.
- Dialogue or speech as main content.
- Poor quality or indistinct segmentation.

# Features of High-Quality Samples：
- Clear event semantics: Each action or change is acoustically distinct and intuitively understandable.
- Unambiguous sequence: Temporal logic is indisputable, allowing only one reconstruction order.
- Narrativity and processuality: The audio presents a complete, coherent event chain consistent with human understanding and commonsense reasoning.

Figure 10: The prompt for our AI-assisted filtering process on temporal tasks.

# Input
You will receive the following three components:
1.Text description: Labels, annotations, or metadata about the audio.
2.Preliminary judgment based on text: A text-only hypothesis about the audio's attributes.
3.Audio file.
Note: Always base your classification primarily on the acoustic features of the audio itself, using text only as supplementary context.

#Output Format (Strictly follow the four-line structure)
Classification label: Sortable Single Event | Sortable Multi-Event | Uncertain | Not Applicable
Event description: For "Sortable Multi-Event," use the format Event A -> Event B -> Event C to describe the strong causal chain. For "Single Event," describe the core dynamic change. If "Not Applicable", briefly summarize the content.
Reasoning: Explain classification basis, focusing on the three standards: semantic clarity, sequence uniqueness, and logical universality. Refer to specific acoustic features where possible.
Score: (Composite evaluation based on the three standards)
- 5 points: Events clear, sequence strong, logic robust, full structure or rich narrative (≥4 events).
- 4 points: Sequence clear, physics/logic reasonable, but semantics somewhat limited (≤3 events).
- 3 points: Some ambiguity, human listeners may also struggle to interpret
- 2 points: Content vague, minimal sequential logic, not recommended for ordering.
- 1 point: Events unclear, structure absent, unusable for ordering tasks.

# Examples：
Input:
Text description: "A ping-pong ball landing."
Preliminary judgment: –
Audio file: Sharp bouncing sounds, gradually weakening, intervals shortening, finally stopping
Output:
Classification label: Sortable Single Event
Event description: Ping-pong ball bouncing, sound intensity and interval decay until stop.
Reasoning:Clear physical energy decay, unique order, limited semantics.
Score: 4

Input:
Text description: "People counting down for New Year, then celebrating with fireworks."
Preliminary judgment: –
Audio file: 10-second countdown → fireworks → cheering
Output:
Classification label: Sortable Multi-Event
Event description: Countdown (10s) -> Fireworks -> Cheering
Reasoning: Three distinct events, unique order, clear logic, strong semantics.
Score: 5

Input:
Text description: "A segment of continuous machine rumble."
Preliminary judgment: –
Audio file: Indistinct machine operating noise
Output:
Classification label: Not Applicable
Event description: Continuous monotonous machine rumble without structural change.
Reasoning: Lacks identifiable dynamic events or temporal structure, unsuitable for ordering evaluation.
Score: 1

Figure 11: The prompt for our AI-assisted filtering process on temporal tasks.

## C  ROBUST EVALUATION

All questions in STAR-BENCH are presented as clear multiple-choice questions with well-formatted options. We adopt classification accuracy as the evaluation metric. To determine the correctness of a response, we employ string matching to extract either the chosen option label (e.g., <A>) or the full text content of the option from the model's output.

Furthermore, we implement a robust evaluation strategy to ensure rigorous and reliable results. For perception and spatial tasks, we adopt the CircularEval method from MM-Bench (Liu et al., 2024b). Specifically, each question is presented to the model $N$ times ($N$ is the number of options), with the option order cyclically rotated in each run to mitigate potential positional biases. For temporal tasks, we conduct three runs per question with different temporal segment orders to evaluate the model's robustness to sequence variations. Note that due to the significant API costs, GPT-4o Audio was evaluated only once per question. This strategy yields two key metrics: Average Accuracy (AA), the

mean accuracy across all evaluation runs, and All-Correct Rate (ACR), the proportion of questions answered correctly in every single run, which serves as a stronger indicator of model reliability.

For models that do not support multi-audio input (only Audio Flamingo 3 and its Think variant among the models we evaluated), we concatenate the audios with a 2-second silence and specify this in the prompt. In contrast, for models that support multiple audio inputs, we feed them sequentially with textual indices.

To establish a human performance baseline, we conduct a human evaluation on a randomly sampled subset of approximately 10% of the data from each task. This evaluation is performed by 10 university students, from whom we explicitly exclude anyone involved in data annotation or with domain-specific expertise, thereby ensuring a general, non-expert perspective.

## D    BREAKDOWN RESULTS

In this section, we present detailed results for perception, temporal reasoning, and spatial reasoning on STAR-BENCH, as shown in Tab. 4, Tab. 5, and Tab. 6.

## E    FURTHER ANALYSIS AND DISCUSSION

### E.1    HIGH OUTPUT INSTABILITY AND CONCENTRATED PREDICTIONS

The reliability of model outputs on our benchmark is notably low, as evidenced by the stark contrast between their Average Accuracy (AA) and All-Correct-Rate (ACR) scores. Even the top-performing model, Gemini 2.5 Pro, exhibits an average drop of 25.01 percentage points from its AA to its ACR. This issue is even more pronounced for the majority of open-source models, which record an ACR near zero. This score indicates a complete failure to maintain consistent predictions under minor input perturbations. For these models, the instability often manifests as a tendency to concentrate predictions on a specific option, suggesting a reliance on superficial biases rather than genuine understanding.

### E.2    ABLATION STUDY ON SPATIAL REASONING.

As shown in Tab. 6, the results reveal a fundamental limitation of LALMs' in spatial perception. The **native input** inherently discards part of the multi-channel information during model preprocessing, which leads to a significant loss of spatial cues that are essential for fine-grained reasoning. On the other hand, the **channel-wise input** explicitly presents each channel with textual instructions, mitigating some of the information loss. Despite this, most existing models are not trained to handle multi-audio inputs. As a result, they consistently struggle to align channel representations and fail to make reliable use of interaural differences. Overall, the pronounced gap between human and model performance highlights that spatial reasoning in audio remains an unsolved challenge, underscoring the need for audio encoders that natively support multi-channel audio input.

## F    CASE STUDY

In this section, we present several case studies of error analysis, including temporal reasoning (Figs. 12 to 17) and spatial reasoning (Fig. 18).

Table 4: Results for the foundational perception task. Each cell reports AA / ACR: Average Accuracy (AA; overall accuracy across all runs) / All-Correct Rate (ACR; proportion of samples that are correct on every run). The best model in each category is shown in **bold**, and the second best is underlined.

| Model | Size | Absolute Perception Range | | | | Relative Discrimination Sensitivity | | | | | | MA (%) |
|---|---|---|---|---|---|---|---|---|---|---|---|---|
| | | Pitch&Loudness | Azimuth | Elevation | Distance | Pitch | Loudness | Duration | Azimuth | Elevation | Distance | |
| Random Guess | — | 25.00/0.39 | 20.00/0.03 | 25.00/0.39 | 25.00/0.39 | 25.00/0.39 | 25.00/0.39 | 25.00/0.39 | 33.33/3.7 | 25.00/0.39 | 25.00/0.39 | 25.33/0.68 |
| Human | — | 98.67/— | 73.33/— | 66.67/— | 70.00/— | 83.33/— | 85.56/— | 83.33/— | 83.33/— | 38.09/— | 73.68/— | 75.60/— |
| SALMONN | 13B | 14.34/0.00 | 25.83/0.63 | 35.76/0.00 | 33.33/0.00 | 31.04/0.00 | 25.00/0.00 | 28.54/0.00 | 31.39/3.89 | 24.15/0.00 | 12.77/0.00 | 26.22/0.45 |
| Audio Flamingo 3 | 8.4B | 37.59/0.00 | 27.92/3.13 | 28.82/0.00 | 32.84/0.00 | 42.50/1.67 | 28.96/0.00 | 34.79/0.00 | 33.90/6.67 | 35.56/0.00 | | 34.15/1.15 |
| Audio Flamingo 3 think | 8.4B | 51.75/6.99 | 8.75/0.00 | 33.33/1.04 | 8.33/0.00 | 36.04/8.33 | 45.63/2.50 | 59.38/38.33 | 41.11/4.17 | 12.29/0.00 | 10.00/0.00 | 30.66/6.14 |
| Qwen2-Audio-Instruct | 8.4B | 35.66/1.40 | 22.50/0.00 | 48.61/10.76 | 12.75/0.98 | 35.63/0.00 | 16.25/0.00 | 26.46/0.00 | 35.00/8.06 | 21.61/1.69 | 23.88/0.00 | 27.84/2.29 |
| DeSTA2.5-Audio | 8.8B | 16.96/0.00 | 21.25/0.42 | 45.49/1.39 | 35.78/1.47 | 11.67/0.00 | 11.25/0.00 | 22.71/0.00 | 33.06/7.78 | 10.59/0.00 | 29.44/0.00 | 23.82/1.11 |
| BAT | 7B | 0.00/0.00 | 26.04/26.04 | 41.67/41.67 | 23.53/23.53 | 0.00/0.00 | 0.00/0.00 | 0.00/0.00 | 37.50/37.50 | 0.00/0.00 | 0.00/0.00 | 12.87/12.87 |
| Phi4-MM | 5.5B | 9.44/0.00 | 24.17/0.00 | 15.97/0.00 | 26.96/0.00 | 24.38/0.00 | 30.00/0.00 | 27.92/0.00 | 36.94/0.00 | 32.62/0.00 | 27.22/0.00 | 25.56/0.00 |
| Kimi-Audio | 7B | 18.71/0.00 | 18.12/0.00 | 38.19/0.00 | 18.13/0.00 | 24.38/0.00 | 32.29/0.00 | 34.17/0.83 | 39.72/3.89 | 25.00/0.85 | 9.44/0.00 | 25.82/0.56 |
| MiDashengLM | 7B | 48.95/33.57 | 20.63/0.00 | 48.26/11.81 | 29.90/0.98 | 40.00/34.17 | 17.08/0.83 | 23.54/7.50 | 34.72/8.61 | 27.12/1.69 | 42.22/6.11 | 33.24/10.53 |
| Step-Audio-2-mini | 7B | 37.59/0.00 | 20.00/0.00 | 31.60/0.69 | 29.41/0.00 | 25.00/0.00 | 29.17/0.00 | 32.29/0.00 | 20.00/0.00 | 25.00/0.00 | | 28.14/0.07 |
| Gemma-3n-E4B-it | 7.5B | 7.18/0.00 | 24.38/4.17 | 25.00/0.00 | 17.65/0.00 | 38.75/0.00 | 8.75/0.00 | 15.00/5.83 | 40.56/1.94 | 23.73/0.00 | 23.33/0.00 | 22.43/1.19 |
| Ming-Lite-Omni-1.5 | 18.9B | 28.67/0.00 | 20.21/0.00 | 27.78/0.35 | 30.39/3.92 | 16.67/16.67 | 16.67/16.67 | 16.67/16.67 | 41.67/0.28 | 32.81/0.00 | 36.11/0.00 | 26.77/5.46 |
| Qwen-2.5-Omni | 7B | 27.45/3.50 | 18.33/0.21 | 27.57/1.47 | 41.67/1.47 | 48.13/35.00 | 39.79/15.00 | 38.33/26.67 | 16.11/0.28 | 11.02/0.00 | 40.56/2.78 | 30.90/8.64 |
| Xiaomi-MiMo-Audio | 7B | 36.71/5.59 | 18.54/19.17 | 48.26/3.82 | 36.27/2.94 | 46.04/24.17 | 36.46/0.83 | 17.70/16.67 | 40.56/2.22 | 20.98/0.00 | 27.78/1.67 | 32.93/7.71 |
| Xiaomi-MiMo-Audio-think | 7B | 43.01/14.69 | 11.67/0.00 | 25.69/0.00 | 39.21/4.90 | 28.13/3.33 | 15.21/1.67 | 22.71/1.67 | 29.44/2.50 | 21.88/0.45 | 32.22/1.67 | 26.92/3.09 |
| MiniCPM-O-v2.6 | 8B | 46.33/8.39 | 24.58/0.21 | 23.26/0.35 | 29.90/0.00 | 38.13/3.33 | 38.96/4.17 | 32.08/3.33 | 37.22/2.78 | 22.10/0.22 | 22.78/0.00 | 31.53/2.28 |
| GPT-4o Audio | — | 45.28/— | 16.67/— | 44.44/— | 3.92/— | 43.33/— | 36.04/— | 46.46/— | 29.58/— | 11.86/— | 40.00/— | 31.76/— |
| Gemini 2.5 Flash | — | 62.59/18.19 | 12.50/0.00 | 18.06/0.35 | 40.69/1.47 | 48.54/21.67 | 40.83/6.67 | 63.13/27.50 | 37.08/9.17 | 25.42/0.85 | 48.33/4.44 | 39.72/9.03 |
| Gemini 2.5 Pro | — | 86.71/62.94 | 25.83/1.25 | 5.88/0.00 | 41.18/5.88 | 63.33/52.50 | 33.75/15.83 | 78.96/68.33 | 37.08/13.75 | 29.24/6.36 | 64.44/12.22 | 46.64/23.91 |

Table 5: Results for the temporal reasoning task. Each cell reports AA / ACR: Average Accuracy (AA; overall accuracy across all runs) / All-Correct Rate (ACR; proportion of samples that are correct on every run). The best model in each category is shown in **bold**, and the second best is underlined.

| Model | Size | Continuous Processes | | Discrete Event Sequences | | | OA (%) |
|---|---|---|---|---|---|---|---|
| | | Object Spatial Motion | In-Situ State Evolution | Tool & Appliance Operation | Daily Scene Scripts | Event-Triggered Consequences | |
| Random Guess | — | 14.29/0.00 | 14.29/0.00 | 14.29/0.00 | 14.29/0.00 | 14.29/0.00 | 14.29/0.00 |
| Human | — | 91.11/— | 88.89/— | 87.88/— | 83.33/— | 83.33/— | 88.00/— |
| SALMONN | 13B | 13.88/0.74 | 16.12/0.00 | 13.56/1.96 | 13.15/1.11 | 12.50/0.89 | 14.15/0.89 |
| Audio Flamingo 3 | 8.4B | 8.55/0.00 | 10.08/0.47 | 8.66/0.98 | 7.22/1.11 | 8.33/3.13 | 8.67/0.67 |
| Audio Flamingo 3 think | 8.4B | 14.37/0.00 | 11.78/0.93 | 15.36/1.47 | 12.96/2.22 | 11.46/0.00 | 13.59/1.00 |
| Qwen2-Audio-Instruct | 8.4B | 12.89/0.00 | 13.80/0.93 | 12.09/0.00 | 12.22/1.11 | 11.46/0.00 | 12.74/0.44 |
| DeSTA2.5-Audio | 8.8B | 16.98/0.37 | 15.97/1.40 | 19.93/1.47 | 15.56/0.56 | 11.46/0.00 | 16.93/0.89 |
| BAT | 7B | 0.00/0.00 | 0.00/0.00 | 0.00/0.00 | 0.00/0.00 | 0.00/0.00 | 0.00/0.00 |
| Phi4-MM | 5.5B | 17.72/0.00 | 15.50/0.47 | 16.34/0.98 | 17.04/3.89 | 20.83/3.13 | 16.85/1.22 |
| Kimi-Audio | 7B | 18.71/1.49 | 21.55/2.33 | 18.63/0.49 | 15.19/2.22 | 14.58/0.00 | 18.52/1.56 |
| MiDashengLM | 7B | 17.10/0.37 | 13.33/0.00 | 17.16/1.96 | 16.67/2.22 | 21.88/0.00 | 16.30/1.00 |
| Step-Audio-2-mini | 7B | 16.11/0.37 | 14.42/0.00 | 15.52/0.00 | 16.30/0.00 | 15.63/0.00 | 15.59/0.11 |
| Gemma-3n-E4B-it | 7.5B | 17.10/0.00 | 16.59/0.00 | 17.81/0.00 | 13.70/0.00 | 20.83/0.00 | 16.59/0.00 |
| Ming-Lite-Omni-1.5 | 18.9B | 17.47/1.12 | 16.59/0.47 | 13.89/0.00 | 17.59/1.11 | 14.58/0.00 | 16.37/0.67 |
| Qwen-2.5-Omni | 7B | 17.10/0.37 | 15.35/0.93 | 19.77/1.47 | 16.48/0.56 | 11.46/0.00 | 16.96/0.78 |
| Xiaomi-MiMo-Audio | 7B | 18.22/0.00 | 18.14/0.47 | 17.16/0.98 | 20.19/2.22 | 26.04/3.13 | 18.63/0.89 |
| Xiaomi-MiMo-Audio-think | 7B | 16.36/0.37 | 17.36/0.47 | 19.93/1.96 | 18.70/2.22 | 19.79/0.00 | 18.00/1.11 |
| MiniCPM-O-v2.6 | 8B | 16.23/0.00 | 14.26/0.93 | 17.48/0.49 | 17.78/0.56 | 14.58/0.00 | 16.30/0.44 |
| GPT-4o Audio | — | 15.61/— | 16.28/— | 24.02/— | 22.78/— | 25.00/— | 19.44/— |
| Gemini 2.5 Flash | — | 30.86/3.35 | 23.41/3.72 | 38.07/12.75 | 30.19/7.22 | 34.38/9.38 | 30.70/6.56 |
| Gemini 2.5 Pro | — | 63.82/38.66 | 43.72/17.67 | 69.77/46.08 | 57.22/38.33 | 48.96/28.13 | 58.52/34.89 |

Table 6: Results for the spatial reasoning task using native and channel-wise audio input. Each cell reports AA / ACR: Average Accuracy (AA; overall accuracy across all runs) / All-Correct Rate (ACR; proportion of samples that are correct on every run). The best model in each category is shown in **bold**, and the second best is underlined.

| Model | Size | Single-Source Static Localization | | Multi-Source Spatial Relation | | Dynamic Trajectory Tracking | | OA (%) | |
|---|---|---|---|---|---|---|---|---|---|
| | | Native Input | Channel-wise Input | Native Input | Channel-wise Input | Native Input | Channel-wise Input | Native Input | Channel-wise Input |
| Random Guess | — | 33.33/3.70 | — | 33.33/3.70 | — | 33.33/3.70 | — | 33.33/3.70 | — |
| Human | — | 70.00/— | — | 80.00/— | — | 77.00/— | — | 73.72/— | — |
| SALMONN | 13B | 26.15/3.18 | 26.62/3.18 | 28.61/4.42 | 29.50/5.31 | 39.94/0.94 | 38.36/0.94 | 29.62/2.99 | 29.75/3.19 |
| Audio Flamingo 3 | 8.4B | 37.22/1.77 | 42.87/2.12 | 38.35/4.42 | 46.31/10.62 | 44.03/4.72 | 46.23/0.94 | 38.91/2.99 | 44.35/3.78 |
| Audio Flamingo 3 think | 8.4B | 35.45/7.42 | 42.87/13.78 | 37.46/23.01 | 46.02/23.01 | 38.05/18.87 | 37.11/19.81 | 36.45/13.35 | 42.36/17.13 |
| Qwen2-Audio-Instruct | 8.4B | 21.32/8.48 | 6.36/1.77 | 24.78/3.54 | 12.09/4.42 | 15.09/0.94 | 11.64/2.83 | 20.78/5.78 | 8.76/2.59 |
| DeSTA2.5-Audio | 8.8B | 23.67/2.83 | 20.38/4.59 | 34.81/9.73 | 41.30/19.47 | 37.74/10.38 | 32.08/21.70 | 29.15/5.98 | 27.56/11.55 |
| BAT | 7B | 0.00/0.00 | 0.00/0.00 | 0.00/0.00 | 0.00/0.00 | 0.00/0.00 | 0.00/0.00 | 0.00/0.00 | 0.00/0.00 |
| Phi4-MM | 5.5B | 33.10/0.35 | 32.63/0.35 | 27.14/0.88 | 29.79/0.88 | 34.28/0.94 | 33.02/0.00 | 32.01/0.59 | 32.07/0.40 |
| Kimi-Audio | 7B | 27.56/3.53 | 16.49/3.53 | 38.94/15.04 | 22.42/8.85 | 44.03/7.55 | 40.25/8.49 | 33.60/6.97 | 22.84/5.77 |
| MiDashengLM | 7B | 43.11/15.19 | 37.22/17.67 | 45.43/23.89 | 42.77/16.81 | 46.23/30.19 | 45.60/21.70 | 44.29/20.32 | 40.24/18.33 |
| Step-Audio-2-mini | 7B | 33.33/0.00 | 33.33/0.00 | 31.27/0.00 | 37.46/0.00 | 37.74/6.38 | 35.22/2.83 | 33.80/1.34 | 34.66/0.60 |
| Gemma-3n-E4B-it | 7.5B | 23.32/1.41 | 28.27/6.01 | 41.89/15.04 | 36.58/7.96 | 33.96/5.66 | 40.57/8.49 | 29.75/5.37 | 32.74/6.97 |
| Ming-Lite-Omni-1.5 | 18.9B | 20.14/6.36 | 34.63/6.01 | 35.10/9.73 | 35.04/9.73 | 38.36/18.87 | 39.94/20.75 | 27.35/9.76 | 35.39/9.96 |
| Qwen-2.5-Omni | 7B | 39.46/7.07 | 36.98/15.19 | 41.30/18.58 | 35.10/15.93 | 27.04/17.92 | 34.59/8.49 | 37.25/11.95 | 36.05/13.94 |
| Xiaomi-MiMo-Audio | 7B | 36.16/0.71 | 41.58/5.65 | 41.30/5.31 | 38.05/4.42 | 45.28/9.43 | 44.34/9.43 | 39.24/3.58 | 41.37/6.17 |
| Xiaomi-MiMo-Audio-think | 7B | 34.28/7.42 | 25.44/2.83 | 44.54/14.16 | 37.76/7.96 | 36.79/7.55 | 27.99/3.77 | 37.12/8.96 | 28.75/4.18 |
| MiniCPM-O-v2.6 | 8B | 29.92/3.18 | 27.92/2.83 | 43.36/11.50 | 39.53/12.39 | 38.36/26.42 | 35.53/17.92 | 34.73/9.96 | 32.14/8.17 |
| GPT-4o Audio | — | 41.81/— | 42.76/— | 43.07/— | 54.87/— | 39.94/— | 42.45/— | 41.70/— | 45.42/— |
| Gemini 2.5 Flash | — | 24.62/4.95 | 40.75/7.42 | 43.07/15.93 | 43.07/17.70 | 22.64/2.83 | 40.57/11.32 | 28.35/6.97 | 41.23/10.56 |
| Gemini 2.5 Pro | — | 40.87/10.95 | 34.98/11.66 | 48.97/25.66 | 49.26/20.35 | 45.28/14.15 | 47.17/7.55 | 43.62/14.94 | 40.77/12.75 |

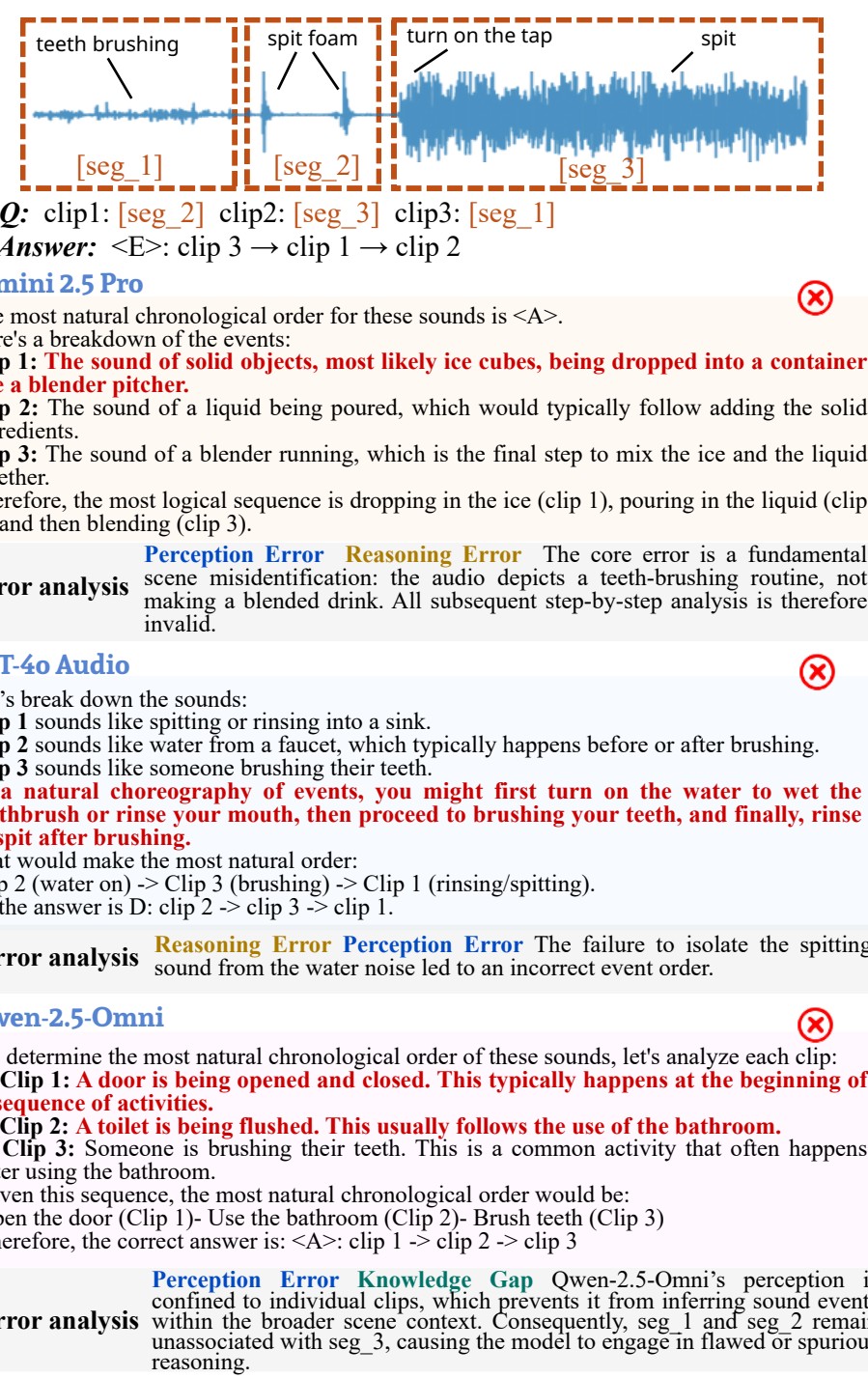

Figure 12: An error case from the temporal reasoning task.

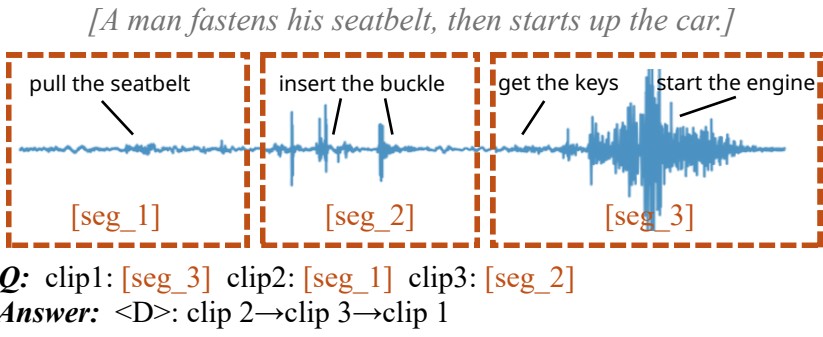

*[A man fastens his seatbelt, then starts up the car.]*

*Q:* clip1: [seg_3]  clip2: [seg_1]  clip3: [seg_2]
*Answer:* <D>: clip 2→clip 3→clip 1

**Gemini 2.5 Pro**

The most natural chronological order for these sounds is <F>. Here's a breakdown of the events:
1. **Clip 3: This is the sound of a car door closing. This is one of the first actions taken after getting into a car.**
2. **Clip 2:** This is the sound of a person settling into the car seat. You can hear clothes rustling and the seat creaking as they get comfortable. This typically happens after the door is closed.
3. **Clip 1:** This is the sound of keys being put into the ignition, followed by the car engine starting. This is the final step in the sequence before driving away.
Therefore, the most logical real-world sequence is closing the door (Clip 3), settling into the seat (Clip 2), and then starting the car (Clip 1).

**Error analysis** **Perception Error** The first half of seg_2 is the sound of pulling the seatbelt, and the second half is the sound of inserting the buckle.

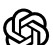 **GPT-4o Audio**

Let me break down the sounds.
**Clip 2** sounds like a zipper, maybe someone zipping up a bag or coat.
**Clip 3** is the sound of a car engine starting, along with some mechanical sounds.
**Clip 1** **sounds like a door closing.**
In a natural sequence, you would expect someone to zip up their bag or jacket, then close a door after stepping out or in, and then finally start the car.
So the most natural chronological order would be: clip 2 (zipping) -> clip 1 (door closing) -> clip 3 (car starting).
That aligns with option E.

**Error analysis** **Perception Error Reasoning-Answer Misalignment** <E> should be clip3→ clip1 → clip2

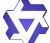 **Qwen-2.5-Omni**

To determine the most natural chronological order of the sounds, let's analyze each clip:
1. **Clip 1:** This clip contains the sound of a car engine starting. This is typically the first action in a sequence of events when someone is about to drive.
2. **Clip 2:** This clip contains the sound of a seatbelt being fastened. This action usually happens after the car engine has started, as the driver or passenger needs to secure themselves before driving.
3. **Clip 3: This clip contains the sound of a car door opening. This action usually happens before the car engine starts, as the driver or passenger needs to enter the car before starting the engine.**
Given this analysis, the most natural chronological order is:
Clip 3: Car door opening - Clip 1: Car engine starting- Clip 2: Seatbelt being fastened
Therefore, the correct answer is:<E>: clip 3 -> clip 1 -> clip 2

**Error analysis** **Perception Error** Seg_3 is the sound of inserting the seatbelt buckle.

Figure 13: An error case from the temporal reasoning task.

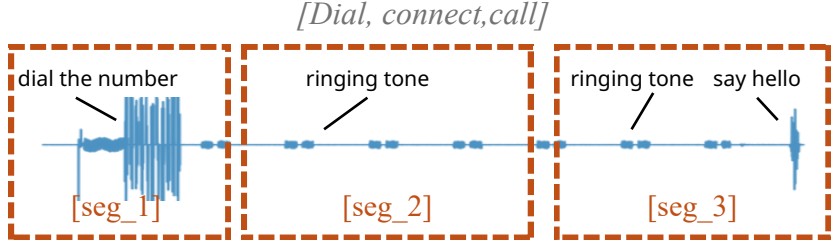

*[Dial, connect,call]*

*Q:* clip1: [seg_2]  clip2: [seg_3]  clip3: [seg_1]
*Answer:* <E>: clip 3→clip 1→clip 2

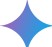 **Gemini 2.5 Pro**

The most natural chronological order for these sounds is:
1. **Clip 3:** The sound of a phone number being dialed on a touch-tone keypad. This is the first action taken to initiate a call.
2. **Clip 1:** The sound of a phone ringing (the ringback tone). This is what the caller hears after dialing and while waiting for the recipient to pick up.
3. **Clip 2:** The sound of someone answering the phone with "Hello?". This is the final step in the sequence, where the call is connected.
Therefore, the correct order is clip 3 -> clip 1 -> clip 2. The correct option is <E>.

**Error analysis**   correct

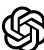 **GPT-4o Audio**

The most natural chronological order for these sounds would typically be:
<A>: clip 1 -> clip 2 -> clip 3
This is because clip 1 sounds like a phone ringing, clip 2 has someone answering the phone with "hello," and **clip 3 is a busy signal**, which can occur after the call is ended or fails to connect.

**Error analysis**   **Perception Error** Seg_3 is the sound of pressing numbers, not a busy signal.

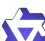 **Qwen-2.5-Omni**

The most natural chronological order in which these sounds would typically occur in the real world is:
<A>: clip 1 -> clip 2 -> clip 3
Explanation:
1. The first sound (clip 1) is a telephone ringing, indicating that someone is calling.
2. The second sound (clip 2) is a person answering the phone, saying "hello."
3. The third sound (clip 3) is **a dial tone, which typically occurs after the call is connected and the call is being made.**

**Error analysis**   **Reasoning Error** Seg_3's explanation is problematic: the dial tone occurs before the call is connected, not after.

Figure 14: An error case from the temporal reasoning task.

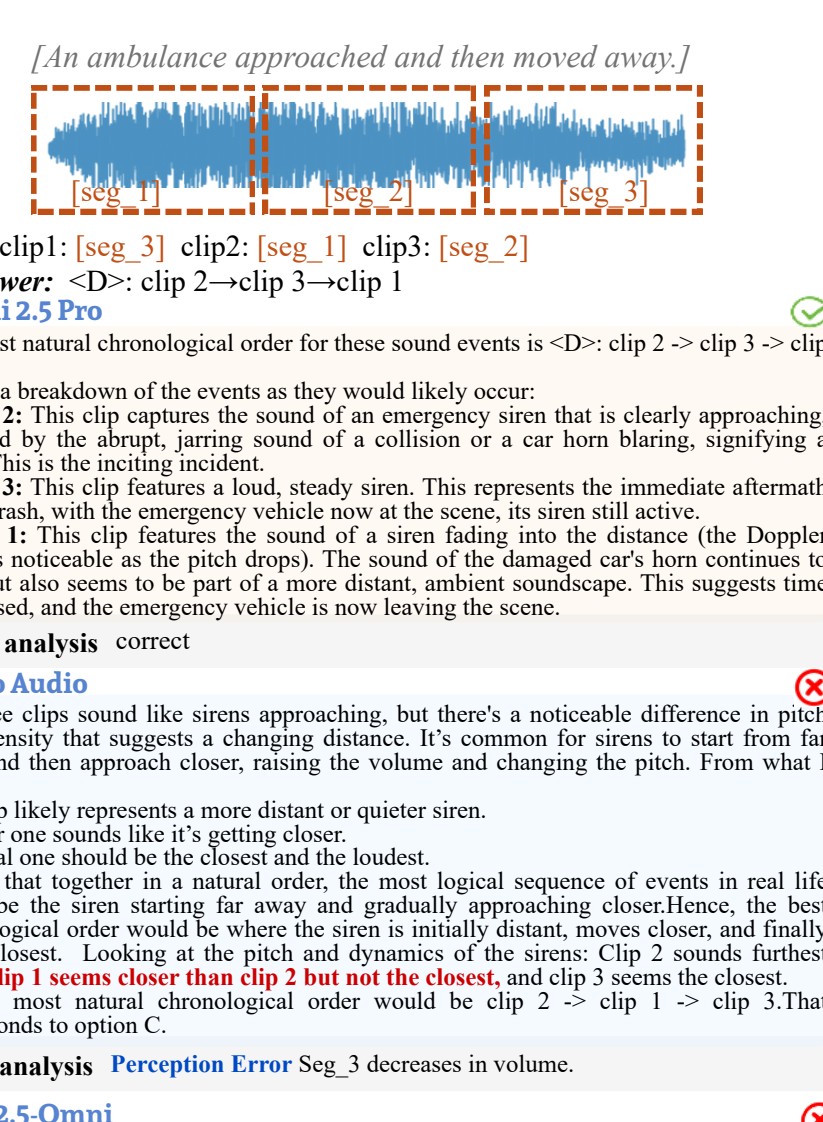

*[An ambulance approached and then moved away.]*

[seg_1]  [seg_2]  [seg_3]

*Q:* clip1: [seg_3]  clip2: [seg_1]  clip3: [seg_2]
*Answer:* <D>: clip 2→clip 3→clip 1

**Gemini 2.5 Pro** ✅

The most natural chronological order for these sound events is <D>: clip 2 -> clip 3 -> clip 1.
Here is a breakdown of the events as they would likely occur:
1. **Clip 2:** This clip captures the sound of an emergency siren that is clearly approaching, followed by the abrupt, jarring sound of a collision or a car horn blaring, signifying a crash. This is the inciting incident.
2. **Clip 3:** This clip features a loud, steady siren. This represents the immediate aftermath of the crash, with the emergency vehicle now at the scene, its siren still active.
3. **Clip 1:** This clip features the sound of a siren fading into the distance (the Doppler effect is noticeable as the pitch drops). The sound of the damaged car's horn continues to blare but also seems to be part of a more distant, ambient soundscape. This suggests time has passed, and the emergency vehicle is now leaving the scene.

**Error analysis**  correct

**GPT-4o Audio** ❌

All three clips sound like sirens approaching, but there's a noticeable difference in pitch and intensity that suggests a changing distance. It's common for sirens to start from far away and then approach closer, raising the volume and changing the pitch. From what I heard:
One clip likely represents a more distant or quieter siren.
Another one sounds like it's getting closer.
The final one should be the closest and the loudest.
Putting that together in a natural order, the most logical sequence of events in real life would be the siren starting far away and gradually approaching closer.Hence, the best chronological order would be where the siren is initially distant, moves closer, and finally is the closest.  Looking at the pitch and dynamics of the sirens: Clip 2 sounds furthest away, **clip 1 seems closer than clip 2 but not the closest,** and clip 3 seems the closest.
So, the most natural chronological order would be clip 2 -> clip 1 -> clip 3.That corresponds to option C.

**Error analysis**  **Perception Error** Seg_3 decreases in volume.

**Qwen-2.5-Omni** ❌

To determine the most natural chronological order of the sounds in the audio clips, let's analyze each clip:
**Clip 1: An emergency vehicle** siren is heard, indicating a situation requiring immediate attention.
**Clip 2: Another emergency vehicle** siren is heard, suggesting a second emergency vehicle is approaching or responding.
**Clip 3: A third emergency vehicle** siren is heard, indicating a third emergency vehicle is also responding.
Given this sequence, the most natural chronological order would be: First, the first emergency vehicle arrives (Clip 1). Then, the second emergency vehicle arrives (Clip 2). Finally, the third emergency vehicle arrives (Clip 3). Therefore, the correct answer is: <A>: clip 1 -> clip 2 -> clip 3

**Error analysis**  **Knowledge Gap** The descriptions disregard fundamental audio characteristics, making no attempt to leverage pitch or volume variations to infer movement, thereby rendering the sequence meaningless.

Figure 15: An error case from the temporal reasoning task.

*[A person opens a jar, quickly gulps the water and then closes the jar.]*

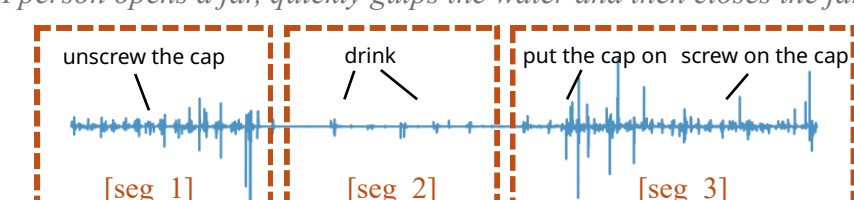

*Q:* clip1: [seg_1]  clip2: [seg_3]  clip3: [seg_2]
*Answer:* : clip 1→clip 3→clip 2

**Gemini 2.5 Pro**

The most natural chronological order for these sounds is <F>: clip 1 -> clip 2 -> clip 3.
Here is a breakdown of the events in that order:
1. **Clip 1:** This clip features the sound of a container, possibly with a powder or solid inside, being shaken. This is followed by the sound of the container being opened (like a cap being twisted off). This is a common preparatory step, like shaking a bottle of drink mix before opening it.
2. **Clip 2:** This clip begins with **the distinct sound of a liquid being poured from a bottle**. This would logically follow opening the container in clip 3, as one would pour a liquid (like water) into the mix. The clip also contains a spraying sound, but the pouring is the key transitional sound from clip 3.
3. **Clip 3:** This clip is the sound of someone drinking and swallowing the liquid. This is the final action of consumption after the drink has been prepared.

**Error analysis**  **Perception Error**  Seg_3 is the sound of closing the lid, not pouring.

**GPT-4o Audio**

Let's break down the sounds logically.
**Clip 1 sounds like ice cubes being placed into a glass.**
**Clip 2 sounds like liquid being poured into that glass.**
Clip 3 sounds like someone sipping or drinking from the glass.
In a natural sequence of events, you would first put ice in the glass, then pour the liquid, and finally drink from it.
Therefore, the most natural chronological order is: clip 1 (ice first), clip 2 (liquid poured next), and clip 3 (drinking the drink last).
That corresponds to option A: clip 1 -> clip 2 -> clip 3.

**Error analysis**  **Perception Error** Descriptions of both seg_1 and seg_3 are incorrect.

**Qwen-2.5-Omni**

To determine the most natural chronological order of the sounds, let's analyze each clip:
1. **Clip 1:** Sounds like someone is stirring or mixing something in a container, possibly a glass or a cup.
2. **Clip 2:** Similar to Clip 1, it also sounds like stirring or mixing, but the sound is slightly different, possibly indicating a different type of container or a different substance.
3. **Clip 3:** Sounds like someone is drinking from a glass or a cup.
Given these observations, the most natural sequence would be:
**1. Stirring or mixing (Clip 1)**
**2. Stirring or mixing (Clip 2)**
3. Drinking (Clip 3)
Therefore, the most natural chronological order is: <A>: clip 1 -> clip 2 -> clip 3

**Error analysis**  **Reasoning Error** Simply placing the two mixing segments together lacks a rational explanation.

Figure 16: An error case from the temporal reasoning task.

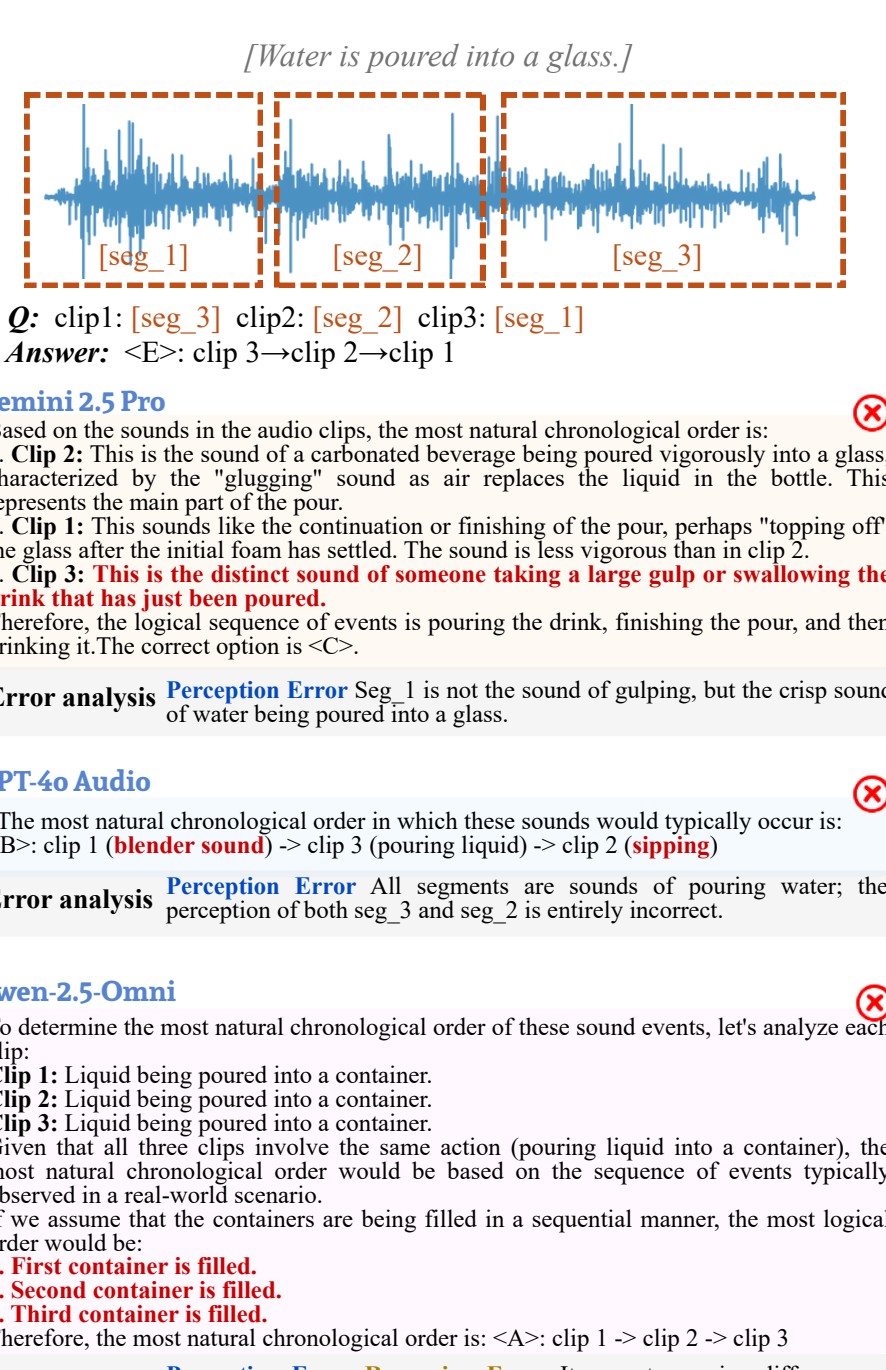

Figure 17: An error case from the temporal reasoning task.

*Q:* What is the movement trajectory of the racing car?
*Options:* <A> From left to right  From right to left <C> Remains unchanged
*Answer:* : From right to left

**native input**

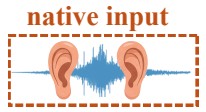

**channel-wise**

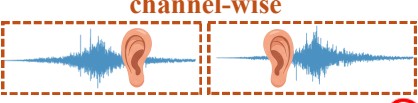

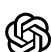 **Gemini 2.5 Pro** ⊗

Based on the audio, the sound of the race car **starts in the left channel, moves across the stereo field, and then fades out in the right channel.** This indicates the car is moving from left to right past the recording point.
Therefore, the correct answer is <A>: From left to right.

**Error analysis**

native: **Perception Error** the channels interfere with each other, causing the model's basic perception to fail
channel-wise: **Perception Error Knowledge Gap** It indicates that the model lacks knowledge for processing spatial audio tasks, relying only on volume to perceive spatial audio, which leads to incorrect answers.

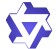 **GPT-4o Audio** ⊗

The racing trajectory of a race car typically involves taking the optimal racing line through each corner to maintain speed and minimize lap time. It often involves starting wide on the approach, cutting close to the apex of the turn, and then moving wide again on the exit, to achieve the smoothest and fastest path. **So, the trajectory changes dynamically depending on the turns and straights of the track.**

**Qwen-2.5-Omni** ⊗

**The audio does not provide information about the direction of the race car's movement, so it is impossible to determine the racing trajectory.** Therefore, the correct answer is C: Remains unchanged.

**Error analysis**

**Perception Error Generation Collapse**
**Hallucination**
It indicates that during the binaural audio processing, the channels interfere with each other, causing the model's basic perception to fail and resulting in hallucinated outputs.

⊗

The correct answer is **<A>: From left to right**.Here's a breakdown of why:
1. **Audio 1 (Left Ear):** The sound of the race car starts, **becomes very loud, and then fades away.**
2. **Audio 2 (Right Ear):** The sound is initially quieter than in the left ear. **It becomes loud as the car passes the center point and then remains louder than the left ear as the car moves away.**
By comparing the two channels, we can hear that the sound is first prominent in the left ear (Audio 1) and then becomes prominent in the right ear (Audio 2). **This change in volume and timing between the ears creates the auditory illusion of the car traveling from the listener's left to their right.**

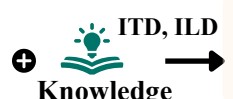 **ITD, ILD** → **Knowledge** ⊕

✓ Based on the binaural cues in the audio recording, the correct answer is **B. From right to left**.

⊗

Based on the audio, the sound of the race car moves from the left channel to the right channel, which means the car is moving from the left to the right side. The correct option is <A>: From left to right.

**Error analysis**

**Hallucination Knowledge Gap**
**Perception Error**
It shown that the model lacks knowledge about spatial audio tasks, making it unable to carry out the correct reasoning process, which in turn leads to hallucinated outputs.

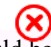
The racing trajectory of a race car would be <A>: From left to right.

Figure 18: An error case from the spatial reasoning task.

