# OpenReview forum: "STAR-Bench: Probing Deep Spatio-Temporal Reasoning as Audio 4D Intelligence"
_ICLR.cc/2026/Conference — ICLR 2026 Poster_

### Official Review · Reviewer_pbHk · 2025-11-01

**Soundness:** 3
**Presentation:** 3
**Contribution:** 3
**Rating:** 6
**Confidence:** 5

**Summary:**

This paper introduces STAR-Bench, a benchmark designed to evaluate “4D Audio Intelligence”, defined as reasoning over sound dynamics in both time (1D) and 3D space. The authors argue that existing audio benchmarks (e.g., MMAU, MMAR) largely assess coarse, text-representable semantics rather than fine-grained perceptual reasoning. STAR-Bench aims to fill this gap through two complementary levels: a) Foundational Acoustic Perception — quantitative evaluation of six acoustic attributes (pitch, loudness, duration, azimuth, elevation, distance) under absolute and relative regimes.b) Holistic Spatio-Temporal Reasoning — tests requiring temporal segment reordering (continuous and discrete processes) and spatial reasoning (localization, multi-source relations, dynamic trajectories). The dataset combines synthetic and real-world audio, curated through a four-stage pipeline involving procedural synthesis, AI-assisted filtering, human annotation, and expert validation. The benchmark comprises 2,353 questions across these tasks.

**Strengths:**

- Establishes a clear and rigorous formalization of “audio 4D intelligence.”
- I agree on the problem formulation.
- The benchmark is novel and challenging. At the same time, it is useful.
- Benchmark design is systematic, combining physical simulation, human validation, and multiple subtask layers.
- Offers a quantitative diagnostic structure — separating perceptual, temporal, and spatial reasoning.
- Provides valuable model insights (e.g., Gemini’s bottleneck in perception vs. open-source deficits in reasoning and grounding).

**Weaknesses:**

- (Minor) The benchmark’s scale (2.3k samples) is relatively small compared to typical multimodal datasets.
- Evaluation focuses mainly on multiple-choice QA
- I don’t have many questions. The work is impressive. Some similar works have popped up lately but can be considered as parallel.
- Some points about Table 1 I am not sure how the ticks and crosses were made. What does Deep Reasoning meaning and why Spatial DR and Temporal DR are crossed for MMAU-Pro? I think spatial is also covered in MMAU-Pro? Can the authors provide more clarity?
- I have no questions. This looks like a well-rounded paper, and I vote for acceptance.

**Questions:**

I don't have questions.

---

> ### Author Response · Authors · 2025-11-20
> **Response to Reviewer pbHk**
>
> We sincerely thank the reviewer for the highly positive assessment of STAR-Bench, including its clear problem formulation, novelty and usefulness, systematic task design, and valuable model insights. We greatly appreciate the reviewer’s thoughtful evaluation. Below, we address the reviewer’s key concerns.
>
> ### **Q1: (Minor) The benchmark’s scale (2.3k samples) is relatively small compared to typical multimodal datasets.**
>   We appreciate your concern. While the benchmark’s scale is relatively small, each sample in STAR-Bench is of **high quality**, and the tasks offer **comprehensive coverage** of audio 4D intelligence. Effective and available data for 4D intelligence reasoning is inherently scarce, so our samples are obtained through multi-stage filtering and high-cost human annotation and validation. In addition, the overall size of STAR-Bench is not smaller than leading benchmarks such as MMAR (1,000 QA pairs) and OmniBench[1] (1,142 samples). We thank the reviewer for the comment and will consider expanding the benchmark further in future iterations.
>
>   [1] Yizhi, L. I., et al. "OmniBench: Towards The Future of Universal Omni-Language Models." The Thirty-ninth Annual Conference on Neural Information Processing Systems Datasets and Benchmarks Track.
>
> ### **Q2:Evaluation focuses mainly on multiple-choice QA**
> Thank you for raising this point.
> Our reliance on multiple-choice questions in this first release is primarily to ensure grading reliability and strict cross-model comparability. In addition, multiple-choice formats remain the dominant design in existing audio benchmarks (e.g., MMAU and MMAR are entirely MCQ-based, and 86.6% of MMAU-Pro questions are MCQ). As an early benchmark specifically targeting audio 4D intelligence, adopting MCQ allows us to control evaluation noise and isolate perceptual and spatio-temporal reasoning abilities, rather than being affected by surface-level linguistic variation—especially given that most open-source models still exhibit limited instruction-following and free-form answering consistency.
> Nevertheless, many tasks in STAR-Bench naturally support open-ended phrasing, and we will explore the feasibility of adding an optional open-ended QA protocol and a corresponding grading toolkit in future iterations. We thank the reviewer for highlighting this valuable direction.
>
>
> ### **Q3: Some points about Table 1 I am not sure how the ticks and crosses were made. What does Deep Reasoning meaning and why Spatial DR and Temporal DR are crossed for MMAU-Pro? I think spatial is also covered in MMAU-Pro? Can the authors provide more clarity?**
>
> Thank you for pointing this out. We apologize for the imprecision and have revised Table 1 and the related‑work discussion accordingly.
>
> In our paper, "Deep" is meant to distinguish tasks that **go beyond surface‑level temporal/spatial perception** (e.g., detecting when something happens or where a single source is) and instead require applying physical or causal knowledge,  multi-step reasoning in complex and diverse real-world scenarios, and integrating information from multiple clips or events. Below, we further elaborate on its specific definition of the temporal and spatial domains:
>
> (1) **Temporal Deep Reasoning.** While MMAU, MMAU‑Pro and MMAR do contain temporal questions, they mainly involve identifying the timing or ordering of events  (e.g., when a sound occurs, which event comes first). These are primarily perceptual‑layer tasks. By contrast, our “temporal deep reasoning” tasks require **understanding physical principles or causal dynamics across segments** (e.g., inferring how a process evolves over time or how one event implies another), which cannot be solved by local timing cues alone. For this reason, we keep previous benchmarks as "Not supported" for temporal deep reasoning.
>
> (2) **Spatial Deep Reasoning.** MMAR includes 16 spatial questions and MMAU‑Pro provides 325 binaural spatial QA items. However, their spatial task coverage is still narrow: The spatial items in MMAR are mostly limited to simple “arriving vs. departing’’ cases which require minimal use of stereo cues, or single-source localization tasks. Similarly, MMAU‑Pro, although larger and binaural, largely centers on locating a single sound source. In contrast, STAR‑Bench introduces **a hierarchical structure** spanning (i) single‑source static localization, (ii) multi‑source spatial relations, and (iii) dynamic trajectory tracking **in complex scenes, with an emphasis on stereo‑cue‑based reasoning**. This is reflected in performance gaps: for example, on MMAU‑Pro Qwen‑Omni‑7B scores 41.20 vs. random 21.20, but on STAR‑Bench its gap narrows (37.25 vs. random 33.33), partly because Qwen‑Omni‑7B only processes mono audio (Figure 3) and cannot fully exploit spatial cues, particularly stereo cues. Accordingly, we now mark MMAR and MMAU‑Pro as “Partially supported or limited amount” for spatial deep reasoning in Table 1.

---

> ### Author Response · Authors · 2025-11-28
>
> Dear Reviewer,
>
>  We sincerely appreciate the thoughtful feedback you have provided. Based on your comments, we have prepared a detailed point-by-point response addressing each question and concern.
>
>  As the discussion period has now entered its final week, we would be grateful if you could take a moment to review our clarifications and let us know whether they adequately address your main points.
>
>  Thank you again for your time, effort, and valuable insights.

---

### Official Review · Reviewer_7C2V · 2025-11-01

**Soundness:** 3
**Presentation:** 3
**Contribution:** 3
**Rating:** 6
**Confidence:** 4

**Summary:**

This paper introduces STAR-BENCH, a new benchmark designed to evaluate what the authors term "audio 4D intelligence", the ability to reason about sound dynamics in 3D space and time. The authors argue that existing audio benchmarks primarily test for semantic content that is easily described by text captions, a claim they support with a "caption-only" evaluation experiment where model performance drops only slightly. STAR-BENCH is proposed to fill this gap by focusing on "linguistically hard-to-describe" cues.

**Strengths:**

1. The paper clearly identifies a weakness in existing audio benchmarks. The "caption-only" experiment (Figure 1) provides empirical evidence that current benchmarks often test text-level semantics rather than fine-grained audio perception.
2. The paper's most significant contribution is identifying and proving the insufficiency of standard audio preprocessing in LALMs. The "additive inverse" experiment in Figure 3 is a simple and effective demonstration that a model's spatial reasoning capabilities are non-existent if it just averages channels to mono.
3. The analysis provides clear takeaways for the community, such as the need for models that natively process multi-channel audio and the severe limitations of open-source models in multi-audio grounding (as shown in Figure 9 ablation).

**Weaknesses:**

1. My primary concern is with the "channel-wise input" proposed for spatial reasoning. While this is a clever workaround for the "mono-averaging" problem, it fundamentally changes the task. Instead of evaluating native spatial audio perception, this method tests a model's ability to reason about two separate mono audio streams guided by an explicit textual prompt (e.g., "Audio 1 is the left-ear channel and Audio 2 is the right-ear channel"). This confounds auditory intelligence with text-based reasoning and an understanding of the experimental setup. It does not truly measure the model's ability to process and fuse interaural cues (ITD, ILD) from a single binaural stream. The paper acknowledges models are not trained on multi-audio inputs, but I think this workaround is a significant compromise.
2. The temporal reordering task relies on "strong sequential uniqueness". This is robust for tasks governed by physics (e.g., Doppler effect , fluid dynamics). However, I am less convinced about the "Daily Scene Scripts" and "Event-Triggered Consequences"  subcategories. For example, in the teeth-brushing case study (Figure 12) , the "correct" order is given as brushing -> spit foam -> turn on tap/spit. However, an alternative like GPT-4o Audio's turn on water -> brush -> rinse/spit is just as logical and common. This suggests "logical universality"  is not as high as claimed.
3. The foundational perception tasks use sine waves for non-spatial attributes. While this provides control, its relevance to perceiving pitch or loudness in complex, real-world sounds (the paper's motivation) is unclear. More importantly, the spatial perception tasks are simulated in rooms with simple source sounds ("alarm," "applause," "telephones"). A critical aspect of real-world spatial hearing is localization and tracking in the presence of diffuse background noise and significant reverberation. It is not clear from the paper or appendix if the benchmark tests performance under varying SNRs or reverberation times (RTs), which are standard parameters in auditory scene analysis.
4. Missing ablation on spatial realism: how do models perform if reverberation or stereo cues are removed?
5. Section 5 implies that STAR-Bench measures general “audio reasoning ability,” but all examples rely on synthetic or controlled sounds. Thus, the benchmark may not generalize to naturalistic auditory scenes (e.g., urban soundscapes, conversational dynamics). At least testing transfer performance would strengthen external validity.

**Questions:**

See Weaknesses.

---

> ### Author Response · Authors · 2025-11-20
> **Response to Reviewer 7C2V （1/2）**
>
> We thank the reviewer for the constructive feedback and for acknowledging our analysis of existing benchmark limitations, preprocessing issues in current LALMs, and the insights our benchmark provides to the community. We address your concerns point by point below.
>
> ### **Q1. The use of channel-wise input for spatial reasoning**
> >My primary concern is with the "channel-wise input" proposed for spatial reasoning. While this is a clever workaround for the "mono-averaging" problem, it fundamentally changes the task. Instead of evaluating native spatial audio perception, this method tests a model's ability to reason about two separate mono audio streams guided by an explicit textual prompt (e.g., "Audio 1 is the left-ear channel and Audio 2 is the right-ear channel"). This confounds auditory intelligence with text-based reasoning and an understanding of the experimental setup. It does not truly measure the model's ability to process and fuse interaural cues (ITD, ILD) from a single binaural stream. The paper acknowledges models are not trained on multi-audio inputs, but I think this workaround is a significant compromise.
>
> We fully agree that native binaural audio is the most natural and principled evaluation setting, and all results in our main tables are based on native input. However, as shown in Figure 3, all evaluated models except BAT automatically average the two channels into mono, discarding most stereo cues. As a result, **their “native binaural” performance becomes equivalent to mono-audio evaluation**.
> The channel-wise setting is not designed to replace native binaural evaluation; rather, it serves as **an ablation study to explore whether current models have any spatial capability when the binaural information is preserved at the input**. It is thus an architecture-driven experimental probe constrained by the limitations of current LALMs, not a redefinition of the core spatial task. The weak performance under this setting further underscores the need for future audio encoders that natively support multi-channel audio input.
> We will clarify this more explicitly in the revision to avoid any misunderstanding.
>
> ### **Q2. Strong Sequential Uniqueness and Logical Universality in Temporal Reasoning**
> >The temporal reordering task relies on "strong sequential uniqueness". This is robust for tasks governed by physics (e.g., Doppler effect , fluid dynamics). However, I am less convinced about the "Daily Scene Scripts" and "Event-Triggered Consequences" subcategories. For example, in the teeth-brushing case study (Figure 12) , the "correct" order is given as brushing -> spit foam -> turn on tap/spit. However, an alternative like GPT-4o Audio's turn on water -> brush -> rinse/spit is just as logical and common. This suggests "logical universality" is not as high as claimed.
>
> We appreciate the reviewer’s attention to this issue. Ensuring both strong sequential uniqueness and logical universality is indeed essential for temporal reasoning tasks, and we devoted substantial effort to guaranteeing these properties:
>  1. The annotation guidelines emphasized both sequential uniqueness and logical universality. During the annotation of candidate samples, annotators were required to write descriptive summaries for each segment and explicitly assess these two properties, along with providing a quality rating. Items receiving a quality score below 3 (on a 5-point scale) were excluded.
> 2. Human validation was performed using the same question format as model evaluation. A question was discarded if more than one-third of experts could not determine the correct order.
> 3. General human evaluation yielded 88% accuracy, providing strong evidence that the retained items exhibit high sequential uniqueness and broad logical universality across different human listeners.
>
> Regarding the reviewer’s example (the tooth-brushing case): although several orderings  may sound plausible in textual description, the correct inference depends on **fine-grained auditory details**, which STAR-Bench is designed to test. In this case, the water segment（seg_3） contains a clear spit sound, which unambiguously indicates the final rinse step, giving the sequence a unique temporal order. GPT-4o Audio’s mistake arises precisely because it ignores this subtle acoustic cue—consistent with our goal of evaluating fine-grained perceptual reasoning. This example is included on our [anonymous demo page](https://starbench-anon-demo.netlify.app/) in Error Examples section, and we invite the reviewer to listen to it.

---

> ### Author Response · Authors · 2025-11-20
> **Response to Reviewer 7C2V （2/2）**
>
> ### **Q3. On the misconception that all audio in STAR-Bench is synthetic**
> >w3:The foundational perception tasks use sine waves for non-spatial attributes. While this provides control, its relevance to perceiving pitch or loudness in complex, real-world sounds (the paper's motivation) is unclear. More importantly, the spatial perception tasks are simulated in rooms with simple source sounds ("alarm," "applause," "telephones"). A critical aspect of real-world spatial hearing is localization and tracking in the presence of diffuse background noise and significant reverberation. It is not clear from the paper or appendix if the benchmark tests performance under varying SNRs or reverberation times (RTs), which are standard parameters in auditory scene analysis.
> >
> >w5:Section 5 implies that STAR-Bench measures general “audio reasoning ability,” but all examples rely on synthetic or controlled sounds. Thus, the benchmark may not generalize to naturalistic auditory scenes (e.g., urban soundscapes, conversational dynamics). At least testing transfer performance would strengthen external validity.
>
> We appreciate the reviewer’s comments and would like to clarify a misunderstanding. Only the Foundational Perception tasks (951 QAs) use synthetic audio, which is intentional: controlled stimuli are necessary to precisely probe the basic perceptual ranges and sensitivities required for 4D audio intelligence. In contrast, all audio used in the Holistic Spatio-Temporal Reasoning tasks (900 + 502 QA pairs) comes from real-world recordings, covering diverse and acoustically complex scenes.
> With respect to the design motivation of the foundational perception tasks and the currently weak model performance on them, evaluating robustness under varying SNRs or reverberation times (RTs) is not the most immediate priority. Such robustness tests become more meaningful once models demonstrate stronger baseline spatial competence. Moreover, the holistic reasoning tasks already include audio with substantial real-world noise, reverberation, and environmental variability.
>
> ### **Q4. Missing ablation on spatial realism: how do models perform if reverberation or stereo cues are removed?**
> We appreciate the reviewer raising this point and would like to clarify the following. Reverberation removal is not feasible for real-world recordings (see Q3 on benchmark audio sources), as reverberation is inseparably embedded in the signal and removing it would significantly distort the acoustic scene.
>
> For stereo cues, this ablation already exists in the current paper. **Because most models average binaural audio to mono, the native-input condition corresponds to the “no stereo cues” setting, while the channel-wise configuration provides the complementary “stereo cues preserved” ablation.** Additionally, we have conducted a human evaluation comparing performance on the native binaural audio against a version converted to mono. As shown in the table below, the dramatic drop in human accuracy confirms that our spatial reasoning tasks are fundamentally reliant on stereo cues.
>
> |                  |Localization | Relation | Trajectory |
> |------------------|---------------------|-------------------|---------------------|
> | Human(Native Binaural)  | 70.00                | 80.00              | 77.00                |
> | Human(Mono)            | 33.33                | 38.89              | 40.00                |
> | Random           | 33.33                | 33.33             | 33.33               |

---

> ### Author Response · Authors · 2025-11-28
>
> Dear Reviewer,
>
>  We sincerely appreciate the thoughtful feedback you have provided. Based on your comments, we have prepared a detailed point-by-point response addressing each question and concern.
>
>  As the discussion period has now entered its final week, we would be grateful if you could take a moment to review our clarifications and let us know whether they adequately address your main points.
>
>  Thank you again for your time, effort, and valuable insights.

---

### Official Review · Reviewer_LG4E · 2025-11-02

**Soundness:** 2
**Presentation:** 3
**Contribution:** 2
**Rating:** 4
**Confidence:** 5

**Summary:**

This paper introduces STAR-Bench, a benchmark to evaluate 4D Audio Intelligence - reasoning over time (1D) and 3D space—via two levels: (a) Foundational Acoustic Perception (six attributes: pitch, loudness, duration, azimuth, elevation, distance; tested in absolute and relative regimes) and (b) Holistic Spatio-Temporal Reasoning (temporal segment re-ordering for continuous/discrete processes; spatial reasoning over static localization, multi-source relations, and dynamic trajectories). The dataset has 2,353 questions and is built through a four-stage pipeline (AI-assisted filtering, human annotation, expert validation) drawing on synthetic audio and real-world corpora (e.g., FSD50K, Clotho, STARSS23). Evaluation is multiple-choice with “robust” perturbations (AA/ACR). Results show humans outperform all models (e.g., ~88% temporal), while the best model (Gemini-2.5 Pro) reaches ~49.6% MA, and a larger drop occurs when answering from captions only, supporting the claim that STAR-Bench targets linguistically hard cues.

**Strengths:**

- Clear problem framing ('audio 4D intelligence) and a structured task design that disaggregates perception, temporal reasoning, and spatial reasoning.
- Useful diagnostic split: absolute vs. relative perceptual tests; temporal re-ordering; spatial subtasks (single-source localization, multi-source relations, dynamic trajectories).
- Strong curation pipeline with AI filtering, human annotation, and expert validation; explicit use of public datasets + simulated audio for coverage and control.
- Robustness intent (AA vs. ACR; circular option shuffling; segment order perturbations) is a step forward relative to one-shot MCQ protocols.
- Empirical evidence that 'caption-only' shortcuts collapse on STAR-Bench more than on MMAU/MMAR, supporting the target of non-linguistic cues. (Claim supported in text/figures.)

**Weaknesses:**

- Everything is framed as multiple-choice with string-match grading. No open-ended QAs, while they are more real world.
- A large portion of audio comes from widely used corpora (FSD50K, Clotho, STARSS23); many models likely pretrained on them. The authors argue the task formulation is novel (re-ordering, spatial relations), but clip-level memorization of events/timbres is still possible.
- The paper doesn't quantify inter-rater agreement, item rejection rates, or ambiguity sources.
- In table 1, the authors claim that previous benchmarks like MMAR and MMAU-Pro, do not have spatial reasoning questions, which is wrong. MMAR also has multi-audio questions.
- MMAU, MMAU-Pro and MMAR also contains Temporal Reasoning questions, what does "Deep" signify in the task name in table 1?
- With the recent advancements in Large Audio Language Models where the models are capable of reasoning over all three modalities of audio - sound, speech and music, this benchmark only focuses on sounds which is limiting. Although the authors explicitly frame STAR-Bench as a move away from traditional ASR/captioning style evaluations toward 4D (spatio-temporal) acoustic reasonin - another signal that speech/music tasks per se are out of scope, I strongly feel there can be multiple tasks framed around speech in spatio-temporal setting.

**Questions:**

See the weakness section.

---

> ### Author Response · Authors · 2025-11-20
> **Response to Reviewer LG4E (1/3)**
>
> We sincerely appreciate the reviewer’s recognition of our contributions, including the clear problem framing, structured task design, useful diagnostic splits, strong curation pipeline, robustness evaluation, and the completeness of evidence supporting our claims. We are grateful for your careful review and constructive suggestions. Below, we address each concern in detail.
>
> ### **Q1. Limitation of the problem format (all questions are MCQ)**
> >Everything is framed as multiple-choice with string-match grading. No open-ended QAs, while they are more real world.
>
> Thank you for raising this point.
> We agree that open-ended QA represents a more realistic interaction format. Our reliance on multiple-choice questions in this first release is primarily to ensure grading reliability and strict cross-model comparability. In addition, multiple-choice formats remain the dominant design in existing audio benchmarks (e.g., MMAU and MMAR are entirely MCQ-based, and 86.6% of MMAU-Pro questions are MCQ). As an early benchmark specifically targeting 4D audio intelligence, adopting MCQ allows us to control evaluation noise and isolate perceptual and spatio-temporal reasoning abilities, rather than being affected by surface-level linguistic variation—especially given that most open-source models still exhibit limited instruction-following and free-form answering consistency.
> Nevertheless, many tasks in STAR-Bench naturally support open-ended phrasing, and we will explore the feasibility of adding an optional open-ended QA protocol and a corresponding grading toolkit in future iterations. We thank the reviewer for highlighting this valuable direction.
>
>
> ### **Q2:The audio in STAR-Bench is not out-of-training-domain for evaluated models.**
> >A large portion of audio comes from widely used corpora (FSD50K, Clotho, STARSS23); many models likely pretrained on them. The authors argue the task formulation is novel (re-ordering, spatial relations), but clip-level memorization of events/timbres is still possible.
>
> Thank you for the thoughtful observation. We agree that many of the underlying audio clips in STAR‑Bench originate from widely used corpora such as FSD50K, Clotho, and STARSS23, and that the evaluated models are likely to have seen portions of this data during pre‑training. This is not an accidental artifact but an intentional design choice at this stage: our goal is to probe 4D audio intelligence **under realistic training conditions** and to encourage the community to **reflect on whether current large‑scale training practices actually endow models with spatio–temporal reasoning abilities**.
>
> Crucially, potential clip‑level memorization of events or timbres does **not** translate into strong performance on STAR‑Bench. As we show in Section 4.1, all models perform poorly, and open‑source models in particular are close to random on several reasoning tasks, despite having likely been exposed to raw audio. This suggests that existing training paradigms often centered on clip‑level tagging, QA, or captioning over **linguistically salient cues** (e.g., using FSD50K for sound event recognition) and do not equip models with the abilities needed for audio 4D intelligence.
>
> In contrast, STAR‑Bench tasks are deliberately formulated around audio 4D intelligence, requiring joint advances in fine‑grained acoustic perception, multi‑step reasoning, multi‑audio integration and world knowledge. These capabilities **cannot** be obtained simply by memorizing individual clips or by minor architectural tweaks plus conventional fine‑tuning on current datasets. In this sense, STAR‑Bench is meant not only as an evaluation benchmark but also as **a diagnostic lens on the limitations of today’s training pipelines**. We agree that a future iteration with entirely out‑of‑training‑domain audio would be valuable, and we view the current version as **a necessary first step** that already reveals important, overlooked weaknesses in existing LALMs.

---

> ### Author Response · Authors · 2025-11-20
> **Response to Reviewer LG4E (2/3)**
>
> ### **Q3：The paper doesn't quantify inter-rater agreement, item rejection rates, or ambiguity sources.**
> Thank you for your valuable feedback. We are not entirely certain of the precise expectation for "quantifying inter-rater agreement," but if the reviewer's concern is about the details of the human annotation and quality control process, we are happy to provide a detailed explanation below.
>
> 1. **On Inter-rater Agreement**
>
> We employed a rigorous, multi-stage process to ensure high inter-rater agreement through consensus-building, rather than relying on a single statistical metric (e.g., Cohen's Kappa), which is less suitable for our complex, multi-faceted annotation tasks. The process is as follows:
>
>    - ***Systematic Training***: All annotators received detailed written guidelines and completed a trial annotation of 10 samples, which were reviewed by experts to ensure a unified understanding of the criteria.
>
>    - ***Inter-annotator Cross-validation***:
>      - Initial Annotation: A sample is first annotated by Annotator A.
>         - Annotation content for Temporal Reasoning: Task compliance checks, segment boundary delineation, textual descriptions for sub-clips and the global audio, scene classification, and audio quality scoring.
>         - Annotation content for Spatial Reasoning: Selecting appropriate segments, task classification, and the generation of a question, the correct answer, and distractor options for the multiple-choice format.
>      - Review and Flagging: The sample is then fully reviewed by Annotator B, who flags any inconsistencies with detailed comments and marks the sample as "failed."
>      - Consensus through Negotiation: Annotators A and B then discuss all flagged issues to reach a consensus and apply corrections.
>      - Expert Arbitration: In the cases where a consensus cannot be reached, the sample is escalated to an expert panel for a final decision. If the experts still cannot agree, the sample is discarded.
>
>    - ***Expert Spot-check***: After passing cross-validation, a random 10% of samples undergo a final quality check by experts to ensure consistency and accuracy. Any discovered issues are then sent back for revision.
>
>   Through this process, we ensure that all samples retained in StarBench have achieved a high level of inter-annotationor agreement.
>
> 2. **On Ambiguity Sources**
>
> During the cross-validation discussions, primary sources of ambiguity are as follows:
> - For Temporal Reasoning Tasks:(a) The reasonableness of the segmented clip boundaries. (b) The existence of multiple logically plausible orderings for the segmented clips. (c) Significant discrepancies in audio quality scores.(d) Adherence to formatting and content guidelines for the captions.
> - For Spatial Reasoning Tasks:(a) Whether the spatial perception presented in the audio unambiguously aligns with the annotated answer. (b) Whether the constructed question-answer pair clearly necessitates the use of audio spatial cues for resolution. (c) Potential ambiguity in mapping the event name mentioned in the question to a specific sound in the audio. (d) The appropriate difficulty and plausibility of the distractor options.
>
> 3. **On Item Rejection Rates**
>
> For the Temporal Reasoning Task, using FSD50K as an example:
> - Initial pool: ~50,000 samples
> - After pre-filtering (duration ≥ 6s, energy ≥ 100): 11,155 samples
> - After Gemini 2.5-Pro based filtering: 2,254 samples
> - After initial human annotation: 1,367 samples
> - After cross-validation: 1,116 samples
> - Final set after human performance evaluation: 786 samples
> - This means the final dataset represents only ~1.5% of the original pre-filtered pool, underscoring our strict quality criteria.
>
> For the Spatial Reasoning Task:
> - We first utilized the STARSS23 dataset, which provides synchronized audio, video, and ground-truth object movement trajectories.
>   - Initial pool of potential segments (identified via rule-based methods): ~1,000
>   - After initial human annotation (annotators were given the raw audio-visual data, metadata, and rule-based suggestions): ~700 QA pairs
>   - After cross-validation: 511 QA pairs
>   - Final set after human performance evaluation: 416 QA pairs
> - To expand the dynamic trajectory sub-task, we supplemented it with web-sourced data:
>   - Initial audio pool: 154 samples
>   - After pre-filtering (removing mismatched metadata): 124 samples
>   - After initial human annotation:124 QA pairs
>   - After cross-validation: 90 QA pairs
>   - Final set after human performance evaluation: 86 QA pairs
>
>   In summary, this stringent filtering and annotation process, characterized by a high rejection rate, ensures that every sample in STAR-Bench possesses a clear ground truth and high human consensus. This establishes the benchmark's credibility for accurately assessing model reasoning capabilities. We have clarified these details further in the Appendix and thank the reviewer again for helping us improve the clarity of our paper.

---

> ### Author Response · Authors · 2025-11-20
> **Response to Reviewer LG4E (3/3)**
>
> ### **Q4: Correction of Table 1 and Clarification of “Deep” Temporal and Spatial Reasoning**
> >In table 1, the authors claim that previous benchmarks like MMAR and MMAU-Pro, do not have spatial reasoning questions, which is wrong. MMAR also has multi-audio questions.
> >
> >MMAU, MMAU-Pro and MMAR also contains Temporal Reasoning questions, what does "Deep" signify in the task name in table 1?
> >
> Thank you for pointing this out. We apologize for the imprecision and have revised Table 1 and the related‑work discussion accordingly.
>
> In our paper, "Deep" is meant to distinguish tasks that **go beyond surface‑level temporal/spatial perception** (e.g., detecting when something happens or where a single source is) and instead require applying physical or causal knowledge,  multi-step reasoning in complex and diverse real-world scenarios, and integrating information from multiple clips or events. We have now made this definition explicit in the text. Below, we further elaborate on its specific definition of the temporal and spatial domains:
>
> (1) **Temporal Deep Reasoning.** While MMAU, MMAU‑Pro and MMAR do contain temporal questions, they mainly involve identifying the timing or ordering of events  (e.g., when a sound occurs, which event comes first). These are primarily perceptual‑layer tasks. By contrast, our “temporal deep reasoning” tasks require **understanding physical principles or causal dynamics across segments** (e.g., inferring how a process evolves over time or how one event implies another), which cannot be solved by local timing cues alone. For this reason, we keep previous benchmarks as "Not supported" for temporal deep reasoning.
>
> (2) **Spatial Deep Reasoning.** MMAR includes 16 spatial questions and MMAU‑Pro provides 325 binaural spatial QA items. However, their spatial task coverage is still narrow: The spatial items in MMAR are mostly limited to simple “arriving vs. departing’’ cases which require minimal use of stereo cues, or single-source localization tasks. Similarly, MMAU‑Pro, although larger and binaural, largely centers on locating a single sound source. In contrast, STAR‑Bench introduces **a hierarchical structure** spanning (i) single‑source static localization, (ii) multi‑source spatial relations, and (iii) dynamic trajectory tracking **in complex scenes, with an emphasis on stereo‑cue‑based reasoning**. This is reflected in performance gaps: for example, on MMAU‑Pro Qwen‑Omni‑7B scores 41.20 vs. random 21.20, but on STAR‑Bench its gap narrows (37.25 vs. random 33.33), partly because Qwen‑Omni‑7B only processes mono audio (Figure 3) and cannot fully exploit spatial cues, particularly stereo cues. Accordingly, we now mark MMAR and MMAU‑Pro as “Partially supported or limited amount” for spatial deep reasoning in Table 1.
>
> (3)**Multi‑Audio.** MMAR includes 15 multi‑segment questions, and we therefore mark it as "Partially supported or limited amount" for multi‑audio in Table 1.
>
> We thank the reviewer again for highlighting these issues, and we hope the clarified definitions and revised Table 1 more accurately reflect the distinctions across benchmarks.
>
>
> ### **Q5：Exclusion of speech and music domains**
> >With the recent advancements in Large Audio Language Models where the models are capable of reasoning over all three modalities of audio - sound, speech and music, this benchmark only focuses on sounds which is limiting. Although the authors explicitly frame STAR-Bench as a move away from traditional ASR/captioning style evaluations toward 4D (spatio-temporal) acoustic reasonin - another signal that speech/music tasks per se are out of scope, I strongly feel there can be multiple tasks framed around speech in spatio-temporal setting.
>
> Thank you for raising this point. Our benchmark intentionally focuses on **linguistically hard-to-describe audio cues**, which represent the core challenge in 4D (spatio-temporal) acoustic reasoning. Speech, by contrast, inherently carries explicit linguistic information. Incorporating speech-centered tasks would shift the evaluation toward **ASR performance or text-based contextual reasoning**, rather than the audio-grounded reasoning abilities that STAR-Bench aims to measure. Moreover, speech-based QA data can be constructed relatively easily compared to our carefully designed reasoning tasks, which would run counter to our original motivation for building STAR-Bench.
> Similarly, music with lyrics also introduces a strong linguistic component. Although our dataset contains a small amount of music, it is included only to evaluate music pattern reasoning, not lyric understanding.
> For these reasons, and in alignment with the core design principles of STAR-Bench, we made a deliberate decision to exclude tasks centered on speech or lyric-based music. We believe this choice keeps the benchmark aligned with its primary goal: assessing spatio-temporal acoustic reasoning where linguistic cues do not dominate or overshadow the underlying audio reasoning challenge.

---

> ### Author Response · Authors · 2025-11-28
>
> Dear Reviewer,
>
>  We sincerely appreciate the thoughtful feedback you have provided. Based on your comments, we have prepared a detailed point-by-point response addressing each question and concern.
>
>  As the discussion period has now entered its final week, we would be grateful if you could take a moment to review our clarifications and let us know whether they adequately address your main points.
>
>  Thank you again for your time, effort, and valuable insights.

---

### Official Review · Reviewer_5scV · 2025-11-05

**Soundness:** 3
**Presentation:** 4
**Contribution:** 4
**Rating:** 6
**Confidence:** 4

**Summary:**

The paper introduces STAR-Bench which measures Audio 4D Intelligence (reasoning over sound dynamics in time and 3D space). They combine a Foundational Acoustic Perception setting with a Holistic Spatio-Temporal Reasoning setting that includes segment reordering for continuous and discrete processes and spatial tasks spanning static localization, multi-source relations, and dynamic trajectories. Unlike prior benchmarks where caption-only answering reduces accuracy slightly, STAR-Bench induces far larger drops (-31.5% temporal, -35.2% spatial), evidencing its focus on linguistically hard-to-describe cues. They also provide a comprehensive evaluation of 19 LALMs/OLMs.

**Strengths:**

- Introduces and benchmarks multi-audio segment reordering and stereo spacial reasoning which has been ignored by the previous benchmarks
- Proper coverage of non-spatial attributes (Loudness, Pitch, Duration) and spatial attributes (Azimuth, Elevation, Distance)
- Detailed error analysis on why models dont do well on the proposed benchmark
- Thorough reporting of AA and ACR metrics to test the model's reliability
- Really like the finding "a fundamental inability to effectively compare, ground, and integrate information from multiple audio inputs"

**Weaknesses:**

- missing details of AI-Assisted Automated Filtering. What exactly are we filtering using gemini 2.5 pro
- Spatial data is synthetic and does not represent real world use cases.
- Fig 8 not readable
- Not sure if the baselines support spatial audio. Analysis on those models would not provide beneficial information
- Support for spatial audio in these models can lead to huge jumps in model performance on the benchmark which questions the difficulty of the benchmark

**Questions:**

- Do you think gemini 2.5 pro captioning stage filters the data points to be more biased towards the current capabilities of the models?
- How easy/difficult is it to hill climb on this benchmark for audio models?
- Do the baselines actually support spatial audio?

---

> ### Author Response · Authors · 2025-11-19
> **Response to Reviewer 5scV (1/2)**
>
> We sincerely appreciate the reviewer’s positive assessment of STAR-Bench, including its novelty, proper attribute coverage, detailed analysis, thorough metrics, and insightful findings. Thank you for the thoughtful review and constructive comments. Below, we address each concern in detail.
>
> ### **Q1: Details of AI-Assisted Automated Filtering**
> >missing details of AI-Assisted Automated Filtering. What exactly are we filtering using gemini 2.5 pro?
>
> Thanks for your valuable comment. Gemini 2.5 Pro is used to pre-screen candidate audio clips based on our well-defined taxonomy for temporal reasoning tasks. In Appendix B.3.1 Fig. 10 and Fig. 11 , we provide the full prompt used to query Gemini 2.5 Pro. Briefly, we feed the audio, its metadata, and our task description, and ask Gemini to decide, under our strict criteria of strong sequence uniqueness, semantic clarity, and high logical universality, (i) whether the audio is suitable for a reordering task, (ii) whether it reflects a continuous or discrete process, (iii) the reasoning behind its judgment, and (iv) a quality score. We discard only samples explicitly labeled as “not applicable”; all remaining clips, together with Gemini’s outputs, are then passed to professional annotators for verification and annotation. Thank you for the suggestion. We have revised the paper to make these details clearer.
>
> ### **Q2. Clarification on Synthetic vs. Real-world Spatial Data**
> >Spatial data is synthetic and does not represent real world use cases.
>
>   Sorry for the misunderstanding. To clarify:
> - Only the spatial data used in the Foundational Acoustic Perception setting employs synthetic spatialization, which allows controlled and quantitative evaluation of basic spatial perception.
> - All spatial tasks in the Holistic Reasoning setting are based on real-world spatial audio, recorded in complex environments (including data from STARSS23 and additional real recordings sourced from the internet).
>
>
>
> ### **Q3：Fig 8 not readable**
> Thank you for pointing this out. We have enlarged the figure as much as possible to ensure that the key content is readable in revised paper. If the reviewer wishes to inspect finer details, the figure can be further zoomed in. For convenience, we briefly summarize the main takeaways. The first-row audiograms show that Gemini-2.5-pro achieves a much broader pitch–loudness coverage than GPT-4o audio and Qwen-2.5-Omni (with greener regions indicating higher accuracy and the covered area reflecting perceptual range), while human listeners exhibit near-full coverage. The second-row curves track performance on pitch, loudness, and duration as task difficulty decreases, consistently revealing a stark performance gap between all models and the human baseline. We have added a more detailed explanation in Section 4.2 of the revised paper.
>
> ### **Q4：Spatial Audio Support in Evaluated Models and Its Impact on Benchmark Difficulty**
> >Not sure if the baselines support spatial audio. Analysis on those models would not provide beneficial information
> >
> >Support for spatial audio in these models can lead to huge jumps in model performance on the benchmark which questions the difficulty of the benchmark
> >
> >Do the baselines actually support spatial audio?
>
> Thanks for your valuable comments. As shown in Figure 3, we conducted a simple and effective experiment to test whether the models can process multi-channel spatial audio correctly. **Except for BAT**—which is specifically designed for spatial audio perception—**all other models simply average multi-channel inputs into mono, thus losing most spatial cues**.
>
> Our baseline suite already covers the majority of state‑of‑the‑art LALMs and OLMs. However, **very few existing models natively support spatial audio, and some potentially relevant systems are not yet open‑sourced**. We will be happy to include such models in future experiments as they become available.
>
> Importantly, even BAT, whose encoder is explicitly designed to process binaural audio, performs poorly on STAR‑Bench. This indicates that native multi‑channel support alone does not translate into strong performance: **accepting multi‑channel inputs is not the same as perceiving, grounding, and reasoning over spatial cues**.
>
> Therefore, our results do not support your concern that “support for spatial audio in these models can lead to huge jumps in model performance on the benchmark which questions the difficulty of the benchmark ”. Instead, they suggest that spatial intelligence remains a challenging and underexplored capability, and that **how to effectively support spatial reasoning** in audio models is itself an open research problem.

---

> ### Author Response · Authors · 2025-11-19
> **Response to Reviewer 5scV (2/2)**
>
> ### **Q5: Do you think gemini 2.5 pro captioning stage filters the data points to be more biased towards the current capabilities of the models?**
> We sincerely thank the reviewer for raising this valuable question. We do **not** believe this procedure introduces bias. The reasons are as follows:
>
> - First, the filtering step does not overlap with the core difficulty of our tasks aims to evaluate. The benchmark targets knowledge with high logical universality, and the filtering stage is merely a simple caption-style check used to surface audio clips with clear temporal dynamics from a large corpus. By contrast, the reordering tasks operate on segmented and shuffled audio clips and require fine-grained perception, multi-step reasoning, and cross-segment integration—capabilities far beyond this preliminary triage.
> - Second, the filtering of Gemini is intentionally conservative and is followed by multiple stages of human screening, annotation, and verification. We retain all clips for which Gemini is uncertain and send them to annotators for further review, including additional filtering, segment cutting, cross-validation, and human performance checks. Consequently, the final dataset composition and task difficulty are shaped by our human-designed protocol and labels rather than Gemini’s preferences.
> - Third, our experiments show no evidence of distributional bias that favors Gemini 2.5 pro. Although Gemini 2.5 Pro participates in pre-screening, its performance on STAR-Bench remains far below human level, indicating that the filtering does not confer structural advantages. As detailed in Section 4.2, Gemini’s stronger performance compared to other baselines arises from its intrinsic capabilities—stronger reasoning and knowledge (Figure 6), better perception and temporal grounding (Figures 7 and 8), and superior multi-audio integration—rather than any bias introduced during pre-screening.
>
>
> ### **Q6: How easy/difficult is it to hill climb on this benchmark for audio models?**
> Thanks for raising this valuable question. Meaningful hill climbing on STAR-Bench will require fundamental advancements, not incremental effort, and is therefore challenging. The tasks demand simultaneous improvements in **fine-grained acoustic perception, multi-step reasoning, world knowledge, and multi-audio integration**—capabilities that current LALMs do not easily acquire through simple architectural tweaks or by fine-tuning on existing data formats with conventional training strategies.
>
> For the temporal reasoning tasks, one could, in principle, construct task-specific training data that exactly mirrors our reordering format and fine-tune a model on this novel task to boost scores. However, such effective data is extremely difficult to collect at scale, and even if fine-tuning produces small score gains, these would largely reflect task-format overfitting rather than genuine capability improvement. Notably, we believe none of the evaluated models have been trained on this novel task, yet the performance gap between closed-source and open-source systems remains substantial. This reinforces our view that the reordering task is a novel and meaningful diagnostic for temporal reasoning. Therefore, we do not recommend optimizing models specifically for this task format, as doing so would weaken its intended purpose; this is a sincere suggestion for future users of STAR-Bench.
>
> For spatial intelligence, the challenge is also substantial: encoder architectures that natively support multi-channel spatial audio remain underexplored, and large-scale real-world spatial audio collection and effective learning strategies continue to be open research problems.

---

> ### Author Response · Authors · 2025-11-28
>
> Dear Reviewer,
>
>  We sincerely appreciate the thoughtful feedback you have provided. Based on your comments, we have prepared a detailed point-by-point response addressing each question and concern.
>
>  As the discussion period has now entered its final week, we would be grateful if you could take a moment to review our clarifications and let us know whether they adequately address your main points.
>
>  Thank you again for your time, effort, and valuable insights.

---

### Official Review · Reviewer_M1pg · 2025-11-05

**Soundness:** 3
**Presentation:** 3
**Contribution:** 3
**Rating:** 6
**Confidence:** 4

**Summary:**

A very topical benchmark for large audio language models, which adds basic audio perception modalities, and explores categories where linguistic descriptions are hard.

**Strengths:**

A clear way of generating spatial audio and temporal reasoning benchmarks is described and implemented. The results show clear lack of perception of current LALMs on tasks requiring these skills. This is a valuable and distinct addition to the plethora of benchmarks coming out

**Weaknesses:**

The audio scenes were generated binaurally, but this restricts the range of applications of the benchmark to perhaps humanoid listeners. Also, the HRTF used was a Kemar one, which is somewhat limited; and the scene rendering was also a bit limited. Nonetheless a good beginning. The models are of course set up to fail in this testing.

The spatial audio tasks are relatively hopeless, and analysis of the models was not too much possible To an extent, I would have liked more detailed analysis of the temporal reasoning tasks. An ablation I would like to see is if the models perform better if the  with just single channel presentation of these events to the models.

**Questions:**

How would you generalize the presentation of spatial audio to models, beyond Kemar? How would you accommodate ambisonics? It would have been good to provide some example sounds.

---

> ### Author Response · Authors · 2025-11-20
> **Response to Reviewer M1pg**
>
> We sincerely appreciate your acknowledgement of STAR-Bench as a valuable and distinct addition to current audio benchmarks, as well as your thoughtful and constructive feedback. Below, we address each concern in detail.
>
> ### **Q1: Limitations of binaural presentation of spatial audio**
> >The audio scenes were generated binaurally, but this restricts the range of applications of the benchmark to perhaps humanoid listeners. Also, the HRTF used was a Kemar one, which is somewhat limited; and the scene rendering was also a bit limited. Nonetheless a good beginning. The models are of course set up to fail in this testing.
> >
> >How would you generalize the presentation of spatial audio to models, beyond Kemar? How would you accommodate ambisonics?
>
> We appreciate your insightful comment, which indeed helps us further broaden the range of applications for STAR-Bench.  Our choice of binaural audio in this first release is driven by the current landscape of LALMs: most existing models operate under a **human-like auditory perception assumption** and frequently accept only mono audio inputs. As an early spatial-audio benchmark for LALMs, binaural rendering preserves essential perceptual cues while remaining compatible with today’s architectures.
>
> Regarding the use of a single KEMAR HRTF, we agree that using a single KEMAR HRTF does not cover all possible listener geometries. However, KEMAR is one of the most widely adopted HRTFs in psychoacoustic and spatial audio research, and using a single, standard HRTF helps us control confounding factors and isolate the ability to exploit generic binaural cues (ITD/ILD and spectral shaping), rather than HRTF-idiosyncratic effects. Moreover, the spatial distinctions targeted by our tasks are proven **robust across human listeners**—evidenced both by expert validation (only tasks that passed our expert performance threshold were retained) and by the high accuracy observed in our general human-subject evaluation.
>
> Following your suggestion, we will also include the **corresponding ambisonic audio** in the benchmark release.  All clips in the Foundational Spatial Perception tasks have ambisonics data, and 428 out of 502 clips in the Spatial Reasoning tasks also include FOA; the remaining clips come from online sources without ambisonic versions. This addition will support broader and more flexible spatial-audio evaluation.
>
>
> ### **Q2: Ablation study on single-channel temporal reasoning**
> >The spatial audio tasks are relatively hopeless, and analysis of the models was not too much possible To an extent, I would have liked more detailed analysis of the temporal reasoning tasks. An ablation I would like to see is if the models perform better if the with just single channel presentation of these events to the models.
>
> We apologize for the confusion and would like to clarify that all audio samples related to non-spatial attributes (pitch, loudness, duration) in the foundational perception tasks, as well as all audio samples used in the temporal reasoning tasks, are strictly mono-channel, whereas only the spatial tasks use binaural audio. Therefore, the results of temporal reasoning tasks reported in the paper already correspond to the single-channel ablation you suggested.
>
> Furthermore, as detailed in Section 4.2, we conducted extensive analysis of temporal reasoning, including:
> （1）Error analysis, showing that closed-source models are bottlenecked by fine-grained perception, while open-source models lag across perception, knowledge, and reasoning.
> （2）Ablation studies with stronger references (e.g., +global caption, +uncut audio), revealing that current open-source models struggle to compare, ground, and integrate information across multiple audio inputs.
> If there are additional analyses you are interested in, we would be glad to incorporate them.
>
> ### **Q3: Request for example sounds**
> >It would have been good to provide some example sounds.
>
> We provide an [anonymous demo page](https://starbench-anon-demo.netlify.app/) featuring example audio clips and representative error cases. We invite you to listen with headphones for a more intuitive understanding of the benchmark.

---

> > ### Author Response · Authors · 2025-11-28
> >
> > Dear Reviewer,
> >
> >  We sincerely appreciate the thoughtful feedback you have provided. Based on your comments, we have prepared a detailed point-by-point response addressing each question and concern.
> >
> >  As the discussion period has now entered its final week, we would be grateful if you could take a moment to review our clarifications and let us know whether they adequately address your main points.
> >
> >  Thank you again for your time, effort, and valuable insights.

---

### Author Response · Authors · 2025-11-20
**General Response to All Reviewers**

We sincerely thank all reviewers for their thoughtful reviews and constructive suggestions, which greatly help us refine and strengthen STAR-Bench. We are particularly encouraged by the consistent recognition across most reviews that STAR-Bench brings needed attention to an important and previously underexplored problem in “audio 4D intelligence”, and that our benchmark is regarded as novel, well-motivated, methodologically sound, and valuable to the community. We sincerely appreciate these positive assessments.

## **Key Strengths Highlighted by the Reviewers:**
1. Clear and rigorous formalization of “audio 4D intelligence” and recognition of its importance. *(Reviewer LG4E, Reviewer pbHk)*
2. Identification of limitations in existing benchmarks and focus on linguistically hard-to-describe audio cues, supported by strong empirical evidence. *(Reviewer 5scV, Reviewer LG4E, Reviewer 7C2V, Reviewer pbHk)*
3. A novel, distinct, challenging, and useful benchmark that fills a real gap in the community. *(Reviewer M1pg, Reviewer pbHk)*
4. Systematic task design covering perception, temporal reasoning, and spatial reasoning, with comprehensive and well-structured hierarchical categorization. *(Reviewer 5scV, Reviewer LG4E, Reviewer pbHk)*
5. Strong and rigorous data curation pipeline.*(Reviewer LG4E, Reviewer pbHk)*
6. Robustness evaluation (AA vs. ACR; circular option shuffling; segment order perturbations) that advances beyond one-shot MCQ protocols. *(Reviewer 5scV, Reviewer LG4E)*
7. Detailed error analysis and valuable insights into current model limitations.*(Reviewer 5scV, Reviewer 7C2V, Reviewer pbHk)*


## **Common Updates Across the Rebuttal：**
### 1. Anonymous Demo Page
We now provide an [anonymous demo page](https://starbench-anon-demo.netlify.app/) with representative audio clips and error cases. We invite reviewers to listen with headphones for a more intuitive understanding of STAR-Bench.
### 2. Updated Parts of the Paper (highlighted in blue in the PDF)
The revisions mainly include:
- Corrections to Table 1 and the associated explanations in the Related Work section.
- Adjustment of Fig. 8 to improve readability, together with additional clarifications added in Section 4.2.
- More detailed descriptions of the human annotation procedure in the Appendix.
- Additional details on the Gemini-2.5-Pro filtering process are included in the appendix.

These updates also lead to minor layout adjustments. Please refer to the revised PDF for all changes.
### 3.  Clarification of Audio Types and Sources in STAR-Bench
To address several misunderstandings identified in the reviews regarding STAR-Bench’s composition, we include the table below summarizing the channel configurations and audio sources across different task categories.
|                           | Foundation Perception Tasks                                      |                               | Holistic Reasoning Tasks                 |                       |
|---------------------------|------------------------------------------------------------------|-------------------------------|-------------------------------------------|-----------------------|
| **Category**              | **Non-Spatial Attributes** (Loudness, Pitch, Duration)           | **Spatial Attributes** (Azimuth, Elevation, Distance) | **Temporal Reasoning**            | **Spatial Reasoning** |
| **Channel #**             | mono                                                             | binaural                      | mono                                      | binaural              |
| **Source Type**           | synthetic                                                        | synthetic                    | real-world                                | real-world            |


We again thank all reviewers for their constructive feedback and the  positive recognition of STAR-Bench’s contributions. We address each reviewer’s comments with dedicated point-by-point responses below.

---

### Author Response · Authors · 2025-12-02
**Summary for the Area Chair**

Dear Area Chair：

We sincerely appreciate your efforts in overseeing the review process, especially given the unforeseen challenges and heavy workload. We respectfully submit the following summary. Our initial ratings were **6, 6, 6, 6, and 4**, and we had **not received any reviewer responses** prior to the OpenReview incident. Below, we concisely summarize how we addressed all reviewer concerns.
- Reviewer  M1pg (Rating: 6)
  - Q1: Limitations of binaural presentation
    - We explained that binaural presentation matches current LALM input constraints and has been validated as robust across listeners. The ambisonic version will also be released.
  - Q2: Ablation study on single-channel temporal reasoning
    - All temporal tasks are single-channel, so the reported results already reflect the suggested ablation.
  - Q3: Request for example sounds
    - We provided an [anonymous demo page](https://starbench-anon-demo.netlify.app/) with audio examples.
- Reviewer  5scV (Rating: 6)
  - Q1: Details of AI-Assisted Filtering
    - We added the detailed description to Appendix B.3.1.
  - Q2: Misunderstanding that all spatial data is synthetic
    - We clarified that spatial reasoning tasks use real-world recordings.
  - Q3: Fig 8 not readable
    - We enlarged the figure and added an explanation.
  - Q4: Spatial Audio Support  and Its Impact on Benchmark Difficulty
    - All evaluated models (except BAT) lack native spatial perception. We disagree that spatial-audio support would lead to "huge jumps," as effective spatial reasoning in LALMs remains an open and unsolved problem.
  - Q5: Potential bias introduced by Gemini filtering
    - We explained that the filtering step dose not introduce bias for three reasons: no overlap with core task difficulty, loose pre-screening with strong human checks, and no observed distributional advantage.
  - Q6:  How easy/difficult is it to hill climb on this benchmark?
    - It is difficult. Substantial improvement would require fundamental advances in several inherent capabilities rather than simple architectural tweaks or fine-tuning.
- Reviewer  LG4E (Rating: 4)
  -  Q1: Limitation of MCQ-only format
     - We explained that MCQ is used in this first release to ensure grading reliability and cross-model comparability.
  - Q2: Audio not out-of-training-domain
    -  We clarified that this is an intentional design choice: our goal is to probe 4D audio intelligence under realistic training conditions and to provide a diagnostic lens on the limitations of current training pipelines.
  - Q3: Missing inter-rater agreement, rejection rates, and ambiguity sources
    - We provided detailed descriptions and added these details to Appendix B.3.2.
  - Q4: Correction of Table 1 and Clarification of "Deep" Reasoning
    - We corrected Table 1 (e.g., MMAU-Pro/MMAR deep temporal = "Not supported"; deep spatial ="Partially supported or limited amount"). We clarified that our "deep" temporal reasoning emphasizes physical and causal dynamics across segments, while "deep" spatial reasoning involves richer, stereo-cue-based tasks.
  - Q5: Exclusion of speech and music domains
    - Speech and music (with lyrics) introduce strong linguistic content, shifting evaluation toward ASR and text-based reasoning rather than the linguistically hard-to-describe acoustic cues, and were therefore deliberately excluded.
- Reviewer  7C2V (Rating: 6)
  - Q1: Channel-wise input for spatial reasoning
    - We clarified that native binaural input remains the main evaluation setting, while the channel-wise input serves only as an ablation to artificially preserve stereo cues.
  - Q2: Strong Sequential Uniqueness and Logical Universality in Temporal Reasoning
    - We explained that both properties were ensured through strict annotation guidelines and human validation, supported by 88% general human accuracy. We also noted that the reviewer’s example contains clear audio cues indicating sequential uniqueness.
  - Q3: Misunderstanding that all audio is synthetic.
    - We clarified that all Holistic Spatio-Temporal Reasoning tasks use real-world recordings.
  - Q4: Missing ablation on spatial realism
    - Reverberation cannot be removed from real-world recordings. For stereo cues, the ablation already existed: most models’ native inputs correspond to the "no stereo cues" condition. Human evaluations showed large accuracy drops without stereo cues, underscoring their importance.
- Reviewer  pbHk (Rating: 6)
  - Q1: (minor) Benchmark’s scale
    - While relatively small, STAR-Bench contains high-quality samples with broad coverage of 4D audio intelligence, and its overall size is comparable to benchmarks like MMAR and OmniBench.
  - The other two concerns are identical to Reviewer LG4E’s Q1 and Q4 and are therefore omitted here.

We hope this summary demonstrates that all concerns have been fully addressed, and we appreciate your consideration in the final assessment.

Best regards,

Authors

---

### Meta-Review · Area_Chair_9b6U · 2026-01-06

**Summary:**

Main concerns of this paper includes
- The channel-wise input setting, may confound spatial perception with text-guided multi-audio reasoning;
- whether temporal reordering tasks sufficiently ensure logical uniqueness and avoid ambiguity;
- the relatively small scale of the benchmark and potential in-domain data overlap;
- reliance on MCQ-only evaluation and clarity issues in related work comparisons.

In my opinion,  the third point is an important concern, and is the potential blocker making this benchmark widely used.

**Reviewer Concerns:**

- The channel-wise input setting, may confound spatial perception with text-guided multi-audio reasoning;

Addressed

- whether temporal reordering tasks sufficiently ensure logical uniqueness and avoid ambiguity;

Addressed

- the relatively small scale of the benchmark and potential in-domain data overlap;

Partically, the authors explain the reason, but in my opinion, in-domain data overlap needs more careful check.

- reliance on MCQ-only evaluation and clarity issues in related work comparisons.

Addressed

**Reviewer Scores:**

M1pg
won't change

5scV
6 → 7

LG4E
4 → 5

7C2V
won't change

pbHk
6 → 7

---

### Decision · Program_Chairs · 2026-01-26

Accept (Poster)